

# Strategies for landslide detection using open-access synthetic aperture radar backscatter change in Google Earth Engine

Alexander L. Handwerger[1,2], Shannan Y. Jones[3], Pukar Amatya[4,5,6], Hannah R. Kerner[7], Dalia B. Kirschbaum[6], and Mong-Han Huang[3]

[1]Joint Institute for Regional Earth System Science and Engineering, University of California, Los Angeles, CA, USA
[2]Jet Propulsion Laboratory, California Institute of Technology, Pasadena, CA, USA
[3]Department of Geology, University of Maryland, College Park, MD, USA
[4]Universities Space Research Association, Columbia, MD, USA
[5]Goddard Earth Sciences Technology and Research, Columbia, MD, USA
[6]Hydrological Sciences Laboratory, NASA Goddard Space Flight Center, Greenbelt, MD, USA
[7]Department of Geography, University of Maryland, College Park, MD, USA

*Correspondence to*: Alexander L. Handwerger (alexander.handwerger@jpl.nasa.gov) and Mong-Han Huang (mhhuang@umd.edu)

**Abstract.** Rapid detection of landslides is critical for emergency response, disaster mitigation, and improving our understanding of landslide dynamics. Satellite-based synthetic aperture radar (SAR) can be used to detect landslides, often within days of a triggering event, because it penetrates clouds, operates day and night, and is regularly acquired worldwide. Here we present a SAR backscatter change detection approach that uses multi-temporal stacks of freely available data from the Copernicus Sentinel-1 satellites to detect areas with high landslide density using the cloud-based Google Earth Engine (GEE). Importantly, our approach does not require downloading a large volume of data to a local system or specialized processing software. We provide strategies, including a landslide density heatmap approach, that can aid in rapid response and landslide detection. We test our GEE-based approach on multiple recent rainfall- and earthquake-triggered landslide events. Our ability to detect surface change from landslides generally improves with the total number of SAR images acquired before and after a landslide event, by combining data from both ascending and descending satellite acquisition geometries, and applying topographic masks to remove flat areas unlikely to experience landslides. Importantly, our GEE approach allows the broader hazards and landslide community to utilize and advance these state-of-the-art remote sensing data for improved situational awareness of landslide hazards.

## 1 Introduction

Rapid response to landslide events (and other natural hazards) is necessary to assess damages and save lives. This response effort includes ground-based teams of local residents, government officials and logistics coordinators, scientists, engineers, and more, all working together to identify critically damaged areas (e.g., Benz and Blum, 2019; Inter-Agency Standing Committee, 2015). Yet, many response efforts are impeded by a lack of detailed information on the condition or location of damaged areas following large and widespread landslide events (Lacroix et al., 2018; Robinson et al., 2019). In addition to



rapid response efforts, it is also important to construct accurate landslide inventories in the weeks to months following these events (Froude and Petley, 2018; Roback et al., 2018; Williams et al., 2018). These detailed inventories are used to improve

understanding of where landslides occur, to quantify erosion, and to look for areas where secondary hazards, such as outburst flooding due to landslide dams, may be occurring (Collins and Jibson, 2015; Kirschbaum et al., 2015; Kirschbaum and Stanley, 2018; Roback et al., 2018). It is therefore necessary to develop tools with freely available data that can be used to map the landslide extent and level of damage following catastrophic events.

        Remote sensing techniques are commonly used to construct landslide inventories over large areas following

catastrophic events (e.g., Bessette-Kirton et al., 2019; Roback et al., 2018). Satellite-based optical imagery provides high quality information for landslide mapping. Many studies have leveraged these data with manual (e.g., Harp and Jibson, 1996; Liao and Lee, 2000; Massey et al., 2020) and semi-automated/automated mapping techniques to identify landslides (e.g., Amatya et al., 2019, 2021; Ghorbanzadeh et al., 2019; Hölbling et al., 2015; Lu et al., 2019; Mondini et al., 2011, 2013; Stumpf and Kerle, 2011). While optical imagery provides high quality data, it is often limited in rapid response efforts because optical

imagery requires daylight as well as shadow- and cloud-free conditions for accurately identifying landslides. Persistent cloud cover can prevent landslide mapping from satellite optical imagery for weeks to months (e.g., Lacroix et al., 2018; Robinson et al., 2019).

        Freely available satellite-based synthetic aperture radar (SAR) circumvents some of the optical data limitations because it can penetrate clouds and operate day or night, but SAR data is still limited by geometric shadow and distortion (e.g.,

Adriano et al., 2020; Mondini et al., 2021). SAR data have been used for nearly two decades to investigate landslides (e.g., Colesanti and Wasowski, 2006; Hilley et al., 2004; Roering et al., 2009). However, most studies have focused on interferometric SAR (InSAR), which measures the radar phase change between two acquisitions in order to quantify ground surface deformation (e.g., Handwerger et al., 2019; Huang et al., 2017b; Intrieri et al., 2017; Schlögel et al., 2015).

        SAR radar backscatter intensity- and coherence-based change detection can also be used to detect landslides, floods,

and other types of natural hazards (Burrows et al., 2020; DeVries et al., 2020; Jung and Yun, 2020; Mondini et al., 2019; 2021; Rignot and Van Zyl, 1993; Tay et al., 2020; Yun et al., 2015). Changes in backscatter and coherence occur when there are changes in ground surface properties (e.g., reflectance, roughness, dielectric properties) before and after landslide events. Coherence-based change detection methods work best in urban areas because normally the coherence is high prior to an event, and there is a reduction in coherence from damages after an event. However, backscatter change detection methods can

outperform coherence-based methods in densely vegetated mountainous regions where landslides tend to occur, because in these areas coherence is always low (i.e., no change), while the backscatter can change (see Jung and Yun, 2020 for a detailed comparison between these methods). Currently, SAR-based change detection methods are under-utilized for landslide identification following catastrophic events, which we believe is primarily due to data access and the specialized processing and software required to analyze SAR data.

In this study, we develop tools and strategies for those without SAR expertise to detect individual landslides and areas with high landslide density that can aid in rapid response and in constructing landslide inventories. We define rapid response





landslide detection as the period of time within two weeks following a landslide event (Burrows et al., 2020; Inter-Agency Standing Committee, 2015; Williams et al., 2018). Our tools run in Google Earth Engine (GEE), a free cloud-based online platform, using only freely available data (Gorelick et al., 2017). Recent studies have used GEE to identify floods (e.g., DeVries et al., 2020) and landslides (e.g., Scheip and Wegmann, 2021), investigate land cover (e.g., Huang et al., 2017a) and surface water change (e.g., Donchyts et al., 2016), monitor agriculture (e.g., Dong et al., 2016), and more. For landslide detection, we measure changes in SAR backscatter from the open-access Copernicus Sentinel-1 (S1) satellites. Our approach requires the spatial coordinates of the area of interest (AOI) and the dates and duration of the landslide triggering event. We test our landslide detection tools for recent events occurring under different environmental conditions. To better investigate landslide detection for different types and sizes of landslides, we analyze 1) the 2018 rainfall-induced landslides in Hiroshima Prefecture, Japan, 2) the 2018 earthquake-triggered landslides in Hokkaido, Japan, 3) 2020 rainfall-triggered landslides in Huong Phung, Vietnam, and 4) 2020 rainfall-triggered landslides in Quang Nam, Vietnam (Fig. 1). To determine the most effective strategies for landslide detection, we perform a sensitivity analysis for the 2018 Hiroshima event by varying the time span (i.e., number of images) of SAR data used to measure changes in backscatter before and after the landslide triggering event, and by incorporating topographic slope- and curvature-based masks to remove regions where landslides are unlikely to occur. We then apply our most effective landslide detection strategies to the other case studies. We do not make systematic quantitative comparisons with other remote sensing-based data used for landslide detection (e.g., SAR-based coherence [data not available in GEE] or Normalized Difference Vegetation Index [NDVI]). However, we do make quantitative comparisons of our findings with optical data-based landslide inventories and qualitative comparisons with optical data from the Copernicus Sentinel-2 (S2) satellites for validation of our SAR-based landslide detection. Lastly, we provide application of our landslide detection tools to support rapid response to the August 2021 earthquake-triggered landslide event in Haiti (Fig. 1). The 2021 Haiti event occurred during the writing of this paper and provides the first real time application of our landslide detection approach for rapid response. Our work highlights the utility of using changes in SAR backscatter data to detect areas that have likely experienced landslide activity following catastrophic events, which is necessary for rapid response and for generating landslide inventories in mountainous regions that commonly have persistent cloud cover. Importantly, our GEE approach does not require prior SAR training or software so that the broader hazards communities can utilize these state-of-the-art remote sensing data.

## 2 Methods

The main goals of our work are to provide open-access tools in GEE that will enable users to utilize freely available SAR data to 1) detect areas that have likely experienced landslide activity as fast as possible after triggering events (i.e., rapid detection) and 2) identify landslides for event inventory mapping. We define "detection" and "mapping" using the framework described by Mondini et al. (2021), where detection is "the action of noticing or discovering single or multiple landslide failures in the same general area" and mapping "refers to the action of delineating the geometry of a landslide".



The use of SAR data is particularly important when optical data is limited due to cloud cover. Our methodology is
developed in the GEE "playground" (browser-based graphical user interface) using the JavaScript application programming
interface (API) (Gorelick et al., 2017). This interface allows for coding, mapping/visualization, documentation, and more, and
the products can be easily exported for offline analyses. The GEE codes developed here are published on Github and as shared
GEE script links (see code availability).

## 2.1 SAR Backscatter in Google Earth Engine

We analyzed changes in the SAR backscatter from the S1 satellite constellation to detect ground surface changes associated
with landslides. The S1 constellation currently consists of two satellites, S1-A and S1-B, launched in March 2014 and April
2016, respectively. Each satellite has a minimum 12-day revisit time for a given area. Using data from both satellites provides
a minimum 6-day revisit time. The S1 satellites carry a C-band radar sensor with wavelength ~5.6 cm. Depending on the
location of the AOI, both ascending (asc) and descending (desc) S1 data may be available (see worldwide acquisition coverage,
https://sentinel.esa.int/web/sentinel/missions/sentinel-1/observation-scenario). Currently, S1 is the only SAR data available in
GEE.

GEE provides Level-1 S1 Ground Range Detected (GRD) backscatter intensity coefficient ($\sigma°$) images. The
backscatter coefficient is defined as the target backscattering area (radar cross-section) per unit ground area. All scenes are
processed to remove thermal noise, have undergone radiometric calibration (but not radiometric terrain flattening), and are
orthorectified using the Shuttle Radar Topography Mission (SRTM) Digital Elevation Model (DEM) (Farr et al., 2007), or the
Advanced Spaceborne Thermal Emission and Reflection Radiometer (ASTER) DEM for areas above 60 degrees latitude. All
S1 GRD data values are provided in logarithmic units of decibels (dB) calculated as $10*\log_{10}(\sigma°)$. The backscatter coefficient
is a measure that can be used to determine if the radar signal is scattered towards or away from the SAR sensor. When $\sigma° < 0$,
the dominant scattering is away from the satellite. The direction of scattering is primarily controlled by the geometry of the
landscape relative to the radar look direction and the electromagnetic properties of the land cover. The GEE S1 GRD collection
is updated daily and new data are uploaded to GEE within two days after they become available. GEE ingests all of the
available ascending and/or descending images on-the-fly.

GEE provides GRD images with 10 m pixel spacing and up to four polarization modes: 1) vertical transmit/vertical
receive (VV), 2) horizontal transmit/horizontal receive (HH), 3) vertical transmit/horizontal receive (VV + VH), and 4)
horizontal transmit/vertical receive (HH + HV). HH and HV polarizations are mostly acquired in polar regions, which are thus
unlikely to be useful for landslide detection. For this study we only used SAR data in the VH polarization. Cross-polarizations,
such as VH and HV, are sensitive to forest biomass structure (Le Toan et al., 1992) and are therefore useful to identify
landslides in vegetated areas. We encourage users of our methods to explore the use of other polarizations.



## 2.2 Landslide Detection Approach

Our landslide detection approach requires the user to select an AOI and time period before ($T_{pre}$) and after ($T_{post}$) the Event of Interest (EOI). The AOI can be a single landslide or a mountain range, and the EOI can last from seconds (e.g., earthquakes) to several days (e.g., storms). Thus, it is important to consider the spatial and temporal scale of the EOI when selecting the AOI, $T_{pre}$, and $T_{post}$. Furthermore, $T_{pre}$ and $T_{post}$ will vary depending on the goal of the project (i.e., rapid detection or constructing full event inventories).

To reduce noise from poor quality data, we removed all pixels with values ≤ -30 dB (based on a recommendation from the GEE S1 Data Catalog). We also reduce transient noise and error from atmospheric delay and other sources by stacking images to create pre-event ($I_{pre}$) and post-event ($I_{post}$) backscatter intensity stacks. SAR data stacking has been shown to significantly improve the signal-to-noise ratio in SAR data (e.g., Cavalié et al., 2008; Zebker et al., 1997). Each stack was calculated as the temporal median of the pre-event and post-event SAR data. We constructed image stacks using ascending data, descending data, and combined ascending & descending data. The combined ascending & descending data (also referred to here as "asc & desc") were calculated as the mean of the ascending and descending stacks.

We detected potential landslides by examining the change in the backscatter coefficient using the standard SAR intensity ratio approach, $I_{ratio}$, defined as $I_{ratio} = 10*\log_{10}(I_{pre}/I_{post})$ (e.g., Jung and Yun, 2020; Mondini et al., 2019; 2021). Because GEE provides the backscatter coefficient data in dB, $I_{pre} - I_{post}$ is equivalent to the $I_{ratio}$ (logarithm quotient rule). The $I_{ratio}$ can be either positive or negative, with positive values corresponding to a decrease in the post-event SAR backscatter intensity. SAR backscatter changes following landslide events occur because landslides cause major changes in ground surface properties that alter the radar reflectance, hillslope geometry, roughness, and dielectric properties (Adriano et al., 2020; Mondini et al., 2021; Rignot and Van Zyl, 1993).

We also note that other ground surface change, for instance due to flooding, agriculture, mining, deforestation, and more, can also be detected by examining $I_{ratio}$ (e.g., Jung and Yun, 2020; Tay et al., 2020) and may cause false positives. To help reduce false positives, we removed areas that are unlikely to correspond to landslides (e.g., small lakes, rivers, flat surfaces, hilltops) by using threshold-based masks made from topographic slope and curvature calculated from the 1 arc-second (~30 m) resolution NASADEM, reprocessed SRTM data with improved height accuracy and filled missing elevation data (NASA JPL, 2020). We refer to this mask throughout this paper as the "DEM mask". Since landslides typically initiate on steep hillslopes, slopes less than a few degrees can be masked, as these are the areas that generally correspond to non-landslide locations. However, slope thresholds will vary in different regions. Additionally, it is common for landslides to runout into lower slope areas, therefore it is important to initially consider a wide range of slope values when searching for landslide deposits. We also mask out larger water bodies, including oceans and some lakes and river shorelines using the water body data stored in the NASADEM layer.



## 2.3 SAR Change Detection Performance and Determination of Most Effective Detection Strategies

To determine the performance and most effective strategies of our SAR backscatter change detection for landslides, we performed both quantitative and qualitative comparisons with mapped landslide inventories and satellite optical imagery. We quantitatively evaluated our results with a previously published landslide inventory for the 2018 Hiroshima landslide event using Receiver Operating Characteristic curves (ROC) (Fan et al., 2006). We provide a qualitative visual comparison with published landslide inventories for the 2018 Hiroshima, 2018 Hokkaido, and 2021 Haiti landslide events, and qualitative visual comparisons with S2 optical images for the 2018 Hiroshima, 2018 Hokkaido, and the 2020 Vietnam events. We note that in most cases, especially for rapid response, neither cloud-free optical images nor an external landslide inventory are likely to be available prior to investigation. In some cases, partial cloud-cover optical images can be used to reveal some parts of landslides and help constrain identification of landslides from SAR data. Therefore, we also include S2 imagery in our GEE tools.

We quantified the success of our SAR backscatter change approach to identify true landslides for an extreme rainfall event that caused widespread landsliding in Hiroshima, Japan in 2018. In this case study, we determined the detection performance using ROC, which measures our detection compared to an external landslide inventory under a variety of thresholds for discriminating between landslide and non-landslide pixels. We compared our SAR change detection to the landslide inventory made by the Geospatial Information Authority of Japan (GSI) and Association of Japanese Geographers (AJG) (see data availability). To make a better comparison between our SAR change detection approach and the published landslide inventory, we manually removed landslides with area < 100 m$^2$ from the GSI/AJG inventory for a better match with the minimum size of a 10 m x 10 m S1 SAR pixel. The ROC analyses were performed outside of the GEE platform using the MATLAB software package. We computed the ROC curves for all pixels within the ~277 km$^2$ Hiroshima AOI shown in Figure 2a. For these analyses, each pixel in the SAR intensity change raster was classified as a landslide if the $I_{ratio}$ pixel value was greater than a threshold value, or non-landslide if it was less than the threshold value. The ROC curve is calculated by varying the $I_{ratio}$ threshold values ($I_{TR}$). The initial $I_{TR,ROC}$ is set as the minimum $I_{ratio}$ value in the dataset and is increased until reaching the maximum value (thus we explore the entire range of $I_{ratio}$ values). We then compared these classified pixels to the true landslides in the GSI/AJG inventory. For each threshold, the false positive rate, defined as the ratio of false positives to true non-landslide pixels, is compared to the true positive rate, defined as the ratio of true positives to true landslide pixels. The best performance is determined by maximizing the Area Under the ROC Curve (AUC) (Fan et al., 2006). An AUC = 1 corresponds to a perfect classifier while an AUC = 0.5 is equivalent to a random selection (50% true positive rate and 50% false positive rate). To maximize the AUC, we performed a sensitivity analysis by varying the pre-event and post-event time periods, the satellite acquisition geometry (i.e., asc, desc, or asc & desc), and the thresholds used for slope angle and curvature for the DEM mask. We then apply these lessons learned for optimal landslide detection strategies within the other case studies.

The main goal of our SAR change detection performance analysis was to determine the most effective detection strategies for landslide identification. While we validated and refined our landslide detection approach with external



inventories and optical data in this study, it is important to emphasize that these external data are not required for landslide detection, and our tools are specifically designed to be used without an external landslide inventory and without optical data.

## 2.4 Landslide Density Heatmaps

To identify areas that have likely experienced landslide activity, which is particularly useful for rapid response, we implemented a landslide density heatmap approach. The heatmap is a data visualization technique that consists of a raster made by calculating the density of potential landslide pixels in a location over a given radius using a kernel density estimation. In this way, the landslide density heatmap is a proxy for potential landslide occurrence and not a heatmap of observed landslides classified using other methods. Several recent studies have applied a landslide density map approach to identify critically damaged areas, rather than focus on the location of individual landslides (e.g., Bessette-Kirton et al. 2019; Burrows et al., 2020; Rosi et al., 2018). Our landslide density heatmaps are similar to these other density maps, however instead of counting individual landslides, we calculate the density of individually detected pixels over a fixed area. We define the pixels used in the heatmap by selecting an $I_{ratio}$ threshold for heatmaps ($I_{TR,H}$). We manually explored $I_{TR,H}$ using $I_{ratio}$ percentiles to find the threshold value that visually highlights true landslides and reduces noise and false positives (see Section 4.2 for further explanation). All pixels $\geq I_{TR,H}$ are classified as potential landslide pixels and all pixels below the threshold are excluded from the analyses. By using percentile-based thresholds, we are able to determine thresholds that correspond to landslides in different regions around the world. The $I_{TR,H}$ is different from the thresholds used in the ROC/AUC analyses ($I_{TR,ROC}$) because $I_{TR,H}$ must be defined without the use of an external landslide inventory.

We construct the heatmap using both GEE and QGIS. We use GEE to identify the potential landslide pixel locations and then export these as a KML for heatmap construction in QGIS. Once in QGIS, the pixel locations are converted to a local UTM coordinate system, and then the heatmap is made using the Heatmap (Kernel Density Estimation) processing toolbox. The Heatmap tool box requires selection of a radius, output raster size, and a kernel shape. We found good results with an output pixel size of 100 m and either a quartic or Epanechnikov kernel shape. We did not identify a single best value to define the radius but generally found that radius values between ~1-3 km were appropriate. We encourage users of our tools to explore the heatmap radius as it may vary depending on the AOI and EOI. In addition, we include an option to generate heatmaps directly in GEE.

## 3 Test Sites

### 3.1 2018 Rainfall-triggered Landslides, Hiroshima, Japan

A record-breaking rainfall event occurred between 28 June and 8 July, 2018 in west and central Japan that resulted in widespread floods and landslides. There were more than 200 fatalities, 20,000 damaged buildings and 8,500 damaged houses caused by these natural disasters. Hiroshima Prefecture had an especially high 108 fatalities, 14,862 damaged buildings, and 689 destroyed houses, significantly more than other Prefectures, which was primarily due to the ~8000 triggered landslides



(Adriano et al., 2020; Hirota et al., 2019; Miura, 2019). Between 1–7 July, 2018 there was approximately 500 mm of rainfall
in Hiroshima Prefecture. Cloud cover prevented full landslide detection from optical images during the event period and partial
cloud cover remained for the month following the event.

The study AOI (Fig. 1) has a mixture of land cover including dense forest with little infrastructure in the mountains
and farmlands and residential areas and cities in the valleys. Within our ~277 km$^2$ AOI, there were 3,370 landslides mapped
by the GSI/AJG. The minimum, mean, and maximum elevation is 0, 210, and 850 m, respectively. The mean slope angle is
14° ± 11° (± 1 standard deviation) with a maximum slope of 64°. Adriano et al. (2020) also used SAR intensity change with
data from the ALOS-2 satellite to successfully identify landslides for the same EOI.

### 3.2 2018 Earthquake-triggered Landslides, Hokkaido, Japan

From September 3-5, Typhoon Jebi passed over Japan, which brought about 100 mm of precipitation. A $M_W$ 6.7 earthquake
struck the Iburi-Tobu area of Hokkaido prefecture located in north Japan on 6 September 2018. The powerful ground motion
(station HKD126 records up to 0.67 G; or 153 cm/s PGV) caused liquefaction and triggered ~6,000 landslides, which destroyed
and buried 394 buildings and killed 41 people (Yamagishi and Yamazaki, 2018; Zhang et al., 2019). Most of the coseismic
landslides were classified as coherent shallow debris slides (Zhang et al., 2019). Due to cloud cover, full landslide detection
from optical images was not available until 5 days later. Thus, optical imagery would have also been viable for rapid response
for this event. Nonetheless, we use this EOI to test our GEE SAR-based detection tools.

We defined our study AOI (Fig. 1) as a 1,170 km$^2$ region with a high landslide density (Zhang et al., 2019). The land
cover in our AOI includes dense forests in the mountains and farmlands and cities in the valleys. The minimum, mean, and
maximum elevation is 0, 129, and 625 m respectively. The mean slope angle is 10° ± 8° (± 1 standard deviation) with a
maximum slope of 84°. This EOI was also previously investigated using SAR change detection methods by Adriano et al.,
(2020), Burrows et al. (2020) and Jung and Yun (2020). These studies showed that both coherence change and backscatter
intensity change detection methods work to characterize the landslides, however the backscatter based methods outperform
SAR coherence change detection in the vegetated mountains (Jung and Yun, 2020).

### 3.3 2020 Rainfall-triggered landslides, Huong Phung and Quang Nam, Vietnam

There were several major landslide events in Vietnam in October 2020 that were related to a particularly wet period between
6-28 October as the country was hit by six tropical cyclones (Van Tien et al., 2021a). We examined landslide events from
Huong Phung Commune on 18 October 2020 and Quang Nam Province on 28 October 2020 (Van Tien et al., 2021a). We
selected a 1080 km$^2$ AOI in Huong Phung and a 416 km$^2$ AOI in Quang Nam (Fig. 1). Cloud cover prevented full landslide
detection from optical images during the event period and partial cloud cover remained until February 2021.

The landslides in Huong Phung occurred in an area that had a mean slope angle 17° ± 10° (± 1 standard deviation)
with a maximum slope of 70°. The landslides in Quang Nam occurred in an area that had a mean slope angle 21° ± 9° with a
maximum slope of 71°. Both landslide areas in Vietnam are covered with dense forested vegetation and farmlands.


### 3.4 2021 Earthquake-triggered landslides in Haiti

While writing this paper, a major $M_w$ 7.2 earthquake struck on 14 August 2021 in Haiti and caused widespread landsliding in
the southwestern part of the country. The earthquake epicenter was near Nippes, Haiti (~120 km west of Port-au-Prince)
(https://earthquake.usgs.gov/earthquakes/eventpage/us6000f65h/executive). Two days after the earthquake, heavy rainfall
from Tropical Storm Grace further contributed to the disaster by triggering additional landslides and flooding and hindering
the    earthquake    response    (https://appliedsciences.nasa.gov/what-we-do/disasters/disasters-activations/haiti-earthquake-
landslides-flooding-2021). We were able to test our landslide detection approach in real time to support the response effort.
We used our GEE tools to generate a landslide density heatmap for a ~6500 km$^2$ region.

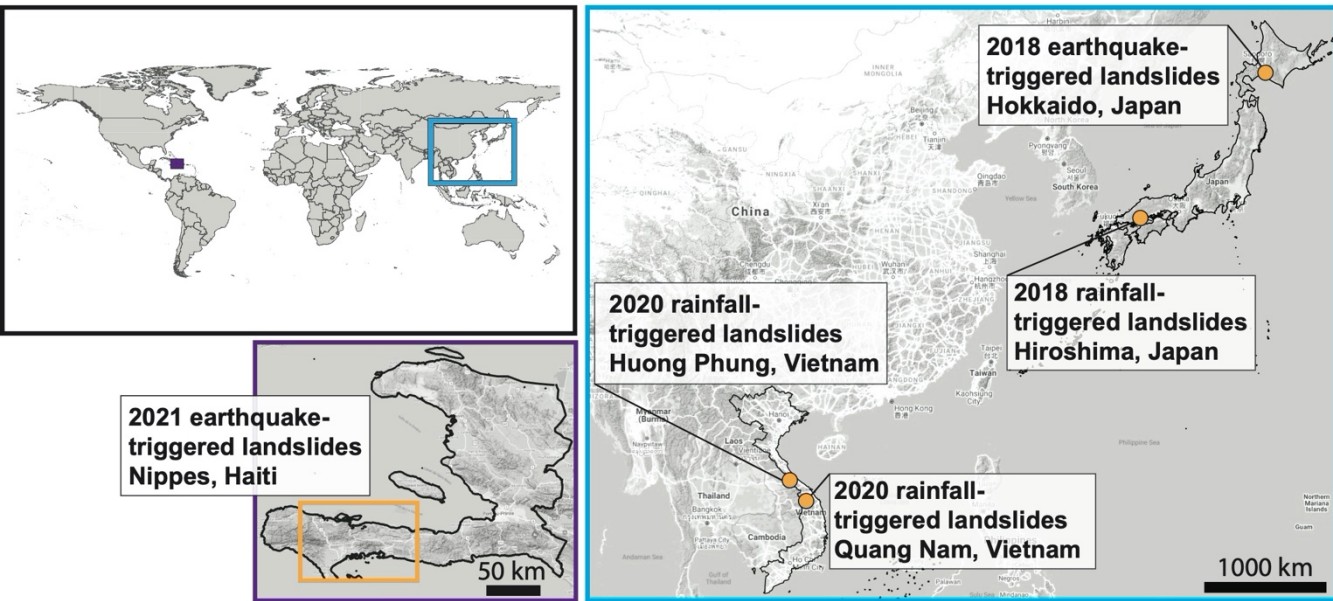

**Figure 1. Google Terrain map showing the location of landslide case studies. © Google Maps 2021**

### 4 Results

We consider five independent landslide events to evaluate our method within Japan, Vietnam, and Haiti. We use the Hiroshima
EOI to determine the most effective landslide detection strategies. We then apply our findings to the other test cases and make
qualitative comparisons with cloud-free optical imagery and published landslide inventories (Hokkaido and Haiti) to help
assess the landslide detection performance.



### 4.1 Determining Effective Strategies for Detecting Landslides

To determine the most effective strategies for landslide detection with SAR backscatter change, and quantify detection success, we explored several different strategies and compared our findings with the GSI/AJG inventory for the 2018 Hiroshima event using the AUC scores computed from the ROC curves. These different strategies included changing the total number of SAR images used in the pre-event and post-event stacks, applying slope and curvature thresholds (i.e., DEM mask), applying $I_{ratio}$ thresholds to highlight landslides and construct heatmaps, and using use ascending, descending, or combined ascending & 280    descending data. We define the "best case" strategy as the approach that maximizes the AUC score.

First, we calculated the SAR backscatter change for the 2018 Hiroshima event using all of the SAR data that was available as of May 29, 2020 (when we began this study) to construct pre- and post-event stacks. The pre-event stack consisted of 142 images (100 ascending and 42 descending) with the first image collected 1,152 days before the EOI and the last image collected 12 days before the EOI. The post-event stack consisted of 133 images (74 ascending and 59 descending) with the 285    first image collected 1 day after the EOI and the last image collected 684 days after the EOI. Figure 2 shows the SAR-based backscatter change for a sub-area of our AOI. The SAR-based backscatter change shows many localized areas with $I_{ratio} > 2$ that correspond to true landslides (Fig. 2c). These relatively high $I_{ratio}$ areas have planform geometries typical of the debris flow type landslides (long, narrow and channelized) that occurred during the 2018 rainfall event. We find most landslides have a positive $I_{ratio}$, i.e., $I_{post} < I_{pre}$, but there were also some places with negative $I_{ratio}$ values within landslide scars as discussed in 290    Adriano et al. (2020). Direct comparison with the cloud-free S2 optical imagery and the GSI/AJG inventory provided initial validation that SAR backscatter change can successfully detect landslides.

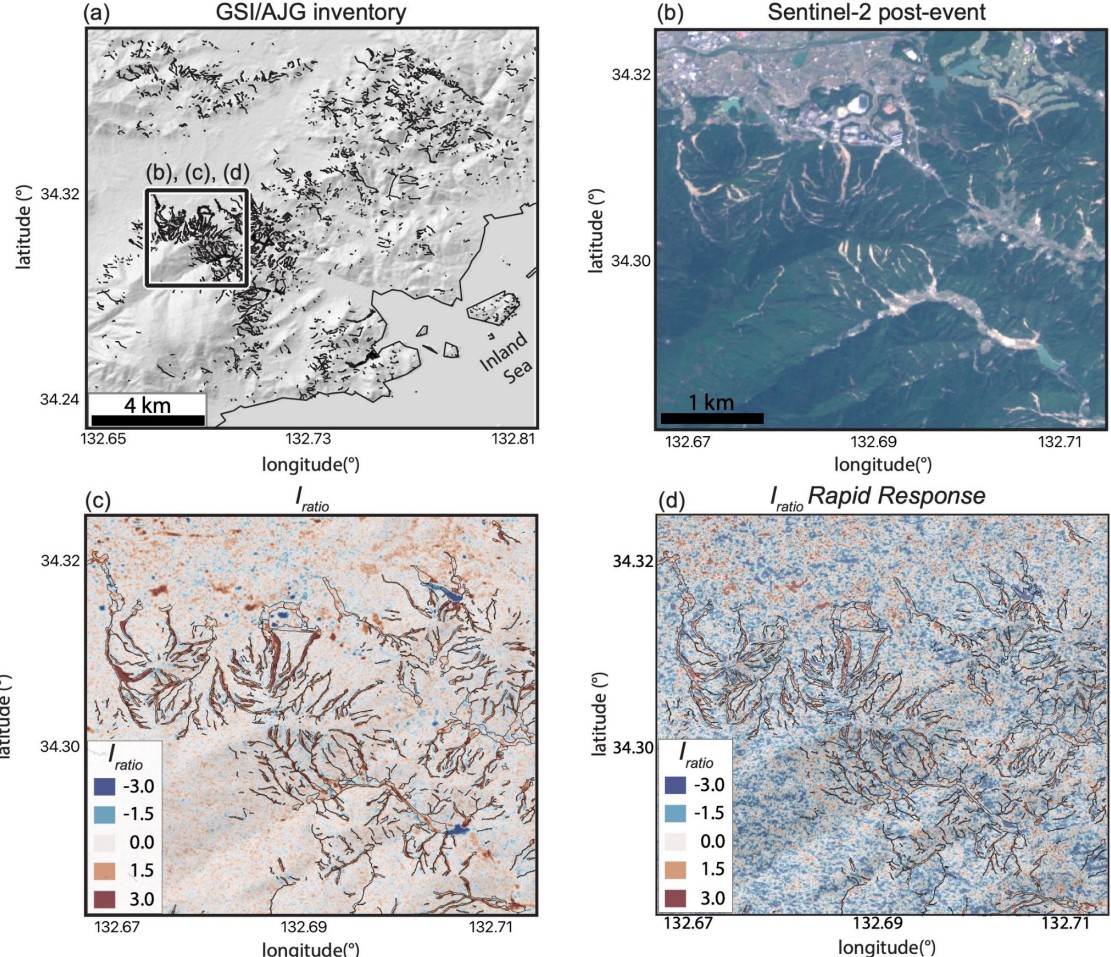

**Figure 2: 2018 rainfall triggered landslides in the Hiroshima Prefecture.** (a) NASADEM hillshade map for Area of interest (AOI) for our study. Black polygons show the GSI/AJG landslide inventory draped on the NASADEM hillshade map with the black box highlighting a sub-area of the AOI for plots b-d. (b) Sentinel-2 optical image showing landslide scars. (c) SAR backscatter intensity change for combined ascending & descending stacks. SAR backscatter change maps were created from stacks made from 142 combined ascending & descending pre-event SAR images collected between 01 May 2015 and 29 June 2018 and 133 post-event SAR images collected between 09 July 2018 and 29 May 2020 and represent our best case landslide identification. (d) SAR backscatter intensity change for simulated rapid response consists of stacks made from the same 142 pre-event SAR images and 5 post-event SAR images collected between 9-24 July 2021. Red colors correspond to a decrease in post-event backscatter intensity (positive $I_{ratio}$ values). Black polygons in (c) and (d) show the GSI/AJG landslide inventory. No DEM mask is applied to (c) or (d).

By comparing our change detection result with the GSI/AJG inventory, we found AUC scores of 0.7363, 0.7409, 0.7712 for ascending, descending, and ascending & descending, respectively using the complete pre- and post-event stacks. We repeated these analyses by changing the time duration for images included in $T_{pre}$ and $T_{post}$ to 12, 6, 3, and 1 months (Fig. 3). We varied the timespan to simulate the impact of different amounts of available data for future studies. The highest AUC





occurred when using all available pre-event and post-event data and by combining asc & desc data. We found that stacking large numbers of SAR images improves the signal-to-noise ratio.

Next we used topographic data to help reduce false positives (Fig. 3b). To determine the slope and curvature thresholds, we used our most effective landslide identification strategy from the previous analyses (i.e., all available pre-event and post-event data) and found the slope and curvature thresholds that maximized the AUC. We found that we can further improve the AUC to 0.8101, 0.8041, 0.8342 ("best case") for asc, desc, and asc & desc, respectively, by using a DEM mask to exclude areas with low topographic slope ($< 5°$) and convex curvature ($> -0.005$ m$^{-1}$). These areas of very low slope

correspond to flat regions, such as cities and valley bottoms, and areas of relatively high positive curvature correspond to hilltops, where landslides are less likely to occur. Importantly, applying slope- and curvature-based masks makes a large improvement in landslide detection, but the specific slope and curvature threshold values will vary for other locations.

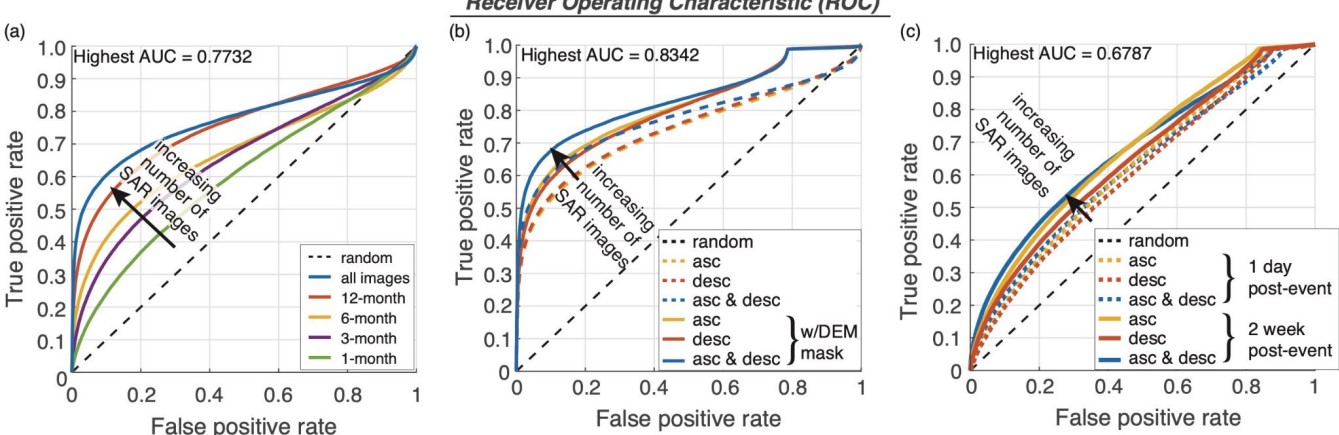

**Figure 3: Receiver Operating Characteristic (ROC) analyses to determine the most effective strategy for landslide identification.** (a) Colored lines show ROC curves as a function of pre-event stack time period ($T_{pre}$) and post-event stack time period ($T_{post}$). (b) ROC curves using all available pre- and post-event data (highest AUC in (a)). Dashed colored lines correspond to ascending (asc), descending (desc), and ascending & descending data. Solid colored lines show the same data with the addition of the DEM mask slope and curvature thresholds. (c) ROC curves with DEM mask for rapid response detection using all pre-event SAR data and post-event data acquired 1 day and 1 week

following the landslide event.

        Lastly, we examined the distribution of $I_{ratio}$ values within the AOI and for true landslides mapped by the GSI/AJG (Fig. A1). The $I_{ratio}$ distribution for the AOI is approximately log normal, but the $I_{ratio}$ for the true landslides is heavy-tailed with positive $I_{ratio}$ values.

## 4.2 Determining Effective Strategies for Rapid Response

To identify landslides for rapid response (i.e., within two weeks following the landslide event) we applied the DEM mask from section 4.1 and explored landslide detection scenarios where we have limited post-event data. Ideally, for rapid response, the first or first few available images following a catastrophic event will provide key information to identify damaged areas. Thus,



our methodology was designed with the goal of being able to provide information to responders on the location of critically damaged areas as quickly as possible.

For the 2018 Hiroshima simulated rapid response, we calculated the SAR backscatter change for a stack consisting of all of the available pre-event imagery and post-event imagery collected within 2 weeks of the EOI. The first post-event images were acquired on July 10, 2018 on both ascending and descending tracks, less than 1 day after the rainfall ended. By making comparison with the GSI/AJG inventory we found an AUC of 0.6212 (Fig. 3c). The second set of post-event images were acquired on July 16, 2018 on both ascending and descending tracks. Incorporating 4 total post-event images improved

the landslide signals and increased the AUC to 0.6787. Examination of the $I_{ratio}$ within the landslides areas shows some elongated debris flow shapes (Fig. 2d), but with a considerably lower signal-to-noise ratio when compared to using post-event stacks with many more images (Fig. 2c). Importantly, the AUC improves rapidly over the first week with the transition from 2 to 4 post-event images, indicating that the images immediately following the event provide key information on the location of damages for rapid response. While the AUC scores are relatively low for the rapid response analyses due to low signal-to-

noise ratio of the post-event images, and it is challenging to identify individual landslides from the $I_{ratio}$, the SAR-based backscatter intensity change still provides key information that can be used to identify the critically damaged areas.

### 4.3 Landslide Detection with SAR-based Change Landslide Density Heatmaps

To rapidly detect areas with high landslide density, we developed a landslide density heatmap approach in GEE. As described above, the heatmap is a data visualization tool that uses the density of potential landslide pixels in a location over a

given radius to generate a raster. A key step in generating a heatmap is to define the $I_{TR,H}$ that classifies true landslide pixels. To select the best $I_{ratio}$ threshold that corresponds primarily to true landslides, we examined data for the 2018 Hiroshima landslide event. We found that $I_{TR,H}$ that includes all pixels ≥ 99th percentile value over the full AOI best highlights the true landslides and removes most of the false positives (Fig. 4). For both the simulated rapid response and the data stack that includes years of post-event data, the 99th percentile highlights true landslides while greatly reducing the false positives.

Comparison with the 90th percentile and 80th percentile thresholds show how the number of false positives increase rapidly by including a larger range of $I_{ratio}$ values. We will use the 99th percentile as our $I_{ratio}$ threshold for the remainder of the study.

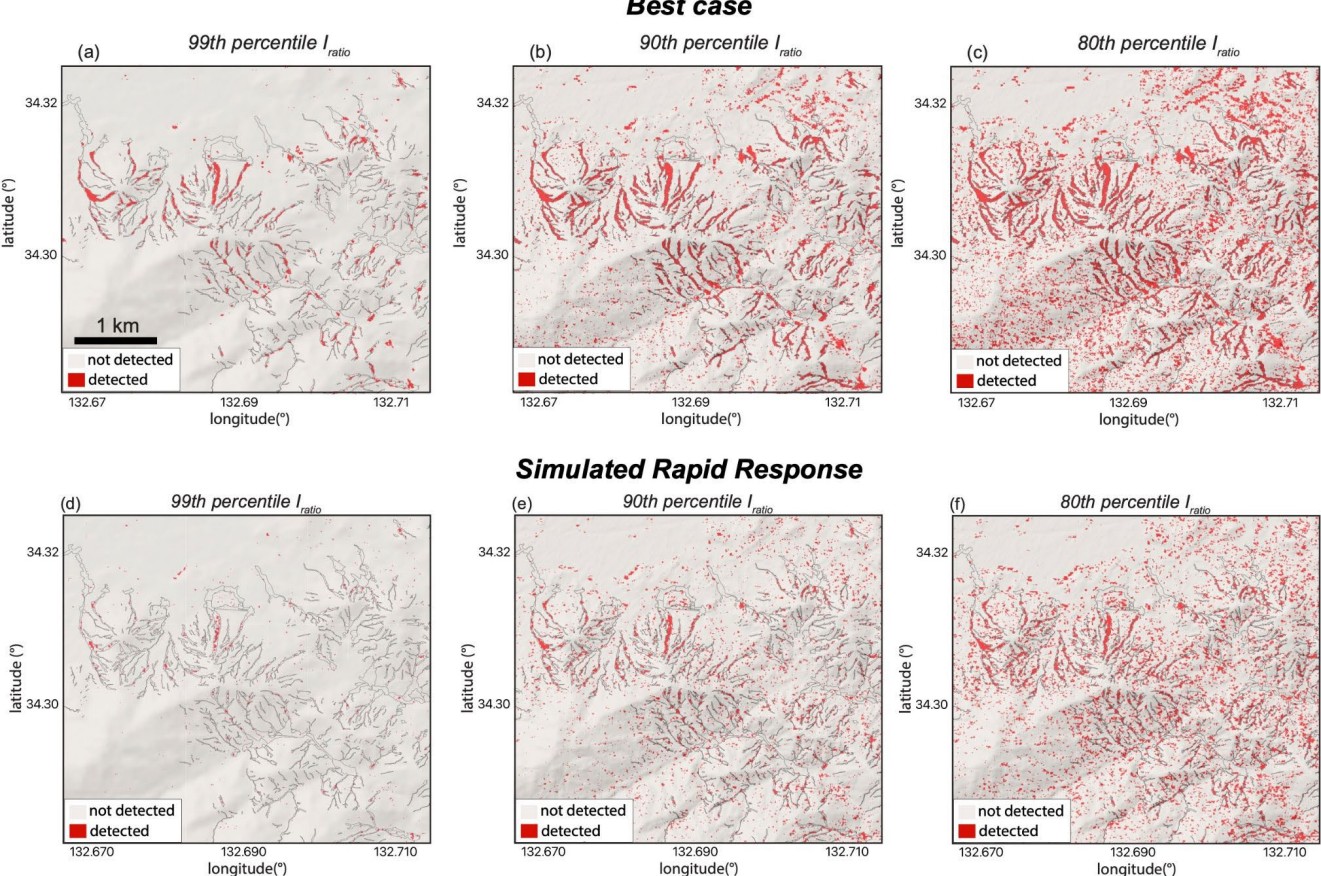

Figure 4: **Detected landslides based on $I_{ratio}$ threshold values ($I_{TR,H}$) for the 2018 Hiroshima landslide event. (**a-c) 99th, 90th, and 80th percentile thresholds for detecting landslides for the "best case" landslide detection strategy that consists of stacks made from 142 combined ascending & descending pre-event SAR images collected between 01 May 2015 and 29 June 2018 and 133 post-event SAR images collected between 09 July 2018 and 29 May 2020. (d-f) 99th, 90th, and 80th percentile thresholds for detecting landslides for the simulated rapid response that consists of stacks made from the same 142 pre-event SAR images and 5 post-event SAR images collected between 9-24 July 2021. DEM mask is used to remove low topographic slope (< 5°) and convex curvature (> -0.005 m$^{-1}$). Thin black polygons show the GSI/AJG landslide inventory.

Figure 5 shows our landslide density heatmap for the simulated rapid response scenario for the 2018 Hiroshima EOI. The heatmap highlights the areas critically damaged by landslides, as shown by comparison with the GSI/AJG inventory, can be located within two weeks of the landslide triggering event. Areas with bright red colors correspond to the areas with highest potential landslide density and/or the largest landslides.

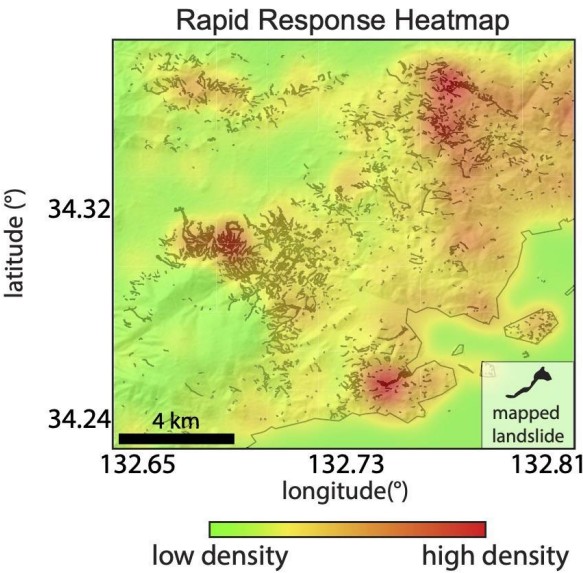

**Figure 5: Simulated rapid response heatmap for the 2018 Hiroshima landslide event.** Heatmap with red colors showing a high density of potential landslides. Black polygons are from GSI/AJG inventory. Pre-event stack created from 142 combined ascending & descending SAR images collected between 01 May 2015 and 29 June 2018 and post-event stack created from 5 SAR images collected between 9-24 July 2021. Heatmap radius set to 100 pixels or 1 km. DEM mask is used to remove low topographic slope (< 5°) and convex curvature (> -0.005 m⁻¹).

### 4.4 Other Case studies

Our analyses of the 2018 Hiroshima event provide us with useful guidelines for landslide detection and rapid response. To further test our approach, we applied our most effective strategies for rapid response to the 2018 earthquake-triggered landslides in Hokkaido, Japan, and 2020 rainfall-triggered landslides in Huong Phung and Quang Nam, Vietnam. Our most effective strategies include using a DEM mask and a large pre-event stack of combined ascending & descending data. We note that the specific values for the DEM mask will vary from location to location so we recommend exploring a range of values. For these case studies, we simulated rapid response scenarios by limiting $T_{post}$ to two weeks of the EOI.

### 4.4.1 Simulated Rapid Response for the 2018 Earthquake-triggered Landslides, Hokkaido, Japan

We defined the simulated rapid response scenario pre-event time period between 1 August 2015 and 5 September 2018 and the post-event time period between 6-21 September 2018 for the Hokkaido event. These time periods resulted in a pre-event stack with 150 combined ascending & descending images and a post-event stack with 3 combined ascending & descending images. We found that the rapid response heatmap characterized the areas with high landslide density (Fig. 6a). Direct visual comparison with the published inventory from Zhang et al. (2019) shows good agreement between the areas detected by our SAR backscatter change heatmap and the true landslides. Figs. 6b-6d show a close up of an area with a particularly high landslide density. The $I_{ratio}$ values and the $I_{ratio} \geq$ 99th percentile show strong and clear signals that correspond to true landslides.





In addition, the $I_{ratio}$ values ≥ 99th percentile appear to correspond almost entirely to true landslides, and examination of the 99th percentile map and heatmap allows for straightforward landslide detection.



**Figure 6. Simulated rapid response heatmap for the 2018 Hokkaido landslide event.** (a) Heatmap with red colors showing a high density detected landslides draped over NASADEM hillshade of topography. Heatmap radius set to 100 pixels or 1 km. (b) Post-event Sentinel-2 optical image for sub-area shown in (a). (c) $I_{ratio}$ map for sub-area. (d) $I_{ratio}$ threshold map with red pixels that are ≥ 99th percentile $I_{ratio}$. Black landslide polygons in (a), (c), and (d) are from Zhang et al., (2019). SAR images made from stacks consisting of 150 pre-event SAR images collected between 01 August 2015 and 05 September 2018 and 3 post-event SAR images collected between 6-21 September 2018. DEM mask is used to remove low topographic slopes < 10° in (a) and (d). No curvature mask is used because the landslides appear to have been sourced from hilltops.

### 4.4.2 Simulated Rapid Response for 2020 Rainfall-triggered Landslides, Huong Phung and Quang Nam, Vietnam

We constructed landslide density heatmaps for simulated rapid response for the October 2020 landslide events in AOIs in Huong Phung (Fig. 7) and Quang Nam (Fig. 8) Vietnam. There were hundreds of landslides triggered during this unusually



wet period, but no published inventory is currently available (to our knowledge). For the Huong Phung event, we defined the
       pre-event period between 1 October 2016 and 17 October 2020 and the post-event period between 19 October 2020 and 1
       November 2020. This resulted in 229 combined ascending & descending pre-event SAR images and 4 post-event SAR images.
       For the Quang Nam event, we defined the pre-event period between 1 October 2016 and 27 October 2020 and the post-event
       period between 29 October 2020 and 7 November 2020. This resulted in 253 combined ascending & descending pre-event
SAR images and 4 post-event SAR images. The heatmaps for both events clearly show areas with high density of detected
       landslides. We checked areas on the heatmaps that showed high landslide density by comparing pre- and post-event S2 optical
       imagery (Figs. 7 and 8). The optical imagery shows the heatmaps successfully highlighted areas with high landslide density.
       We also observed some false positives in the heatmap for the Huong Phung area. We observed apparent high landslide density
       that appeared to correspond to false positives related to clearcutting of trees (Fig. 7e).


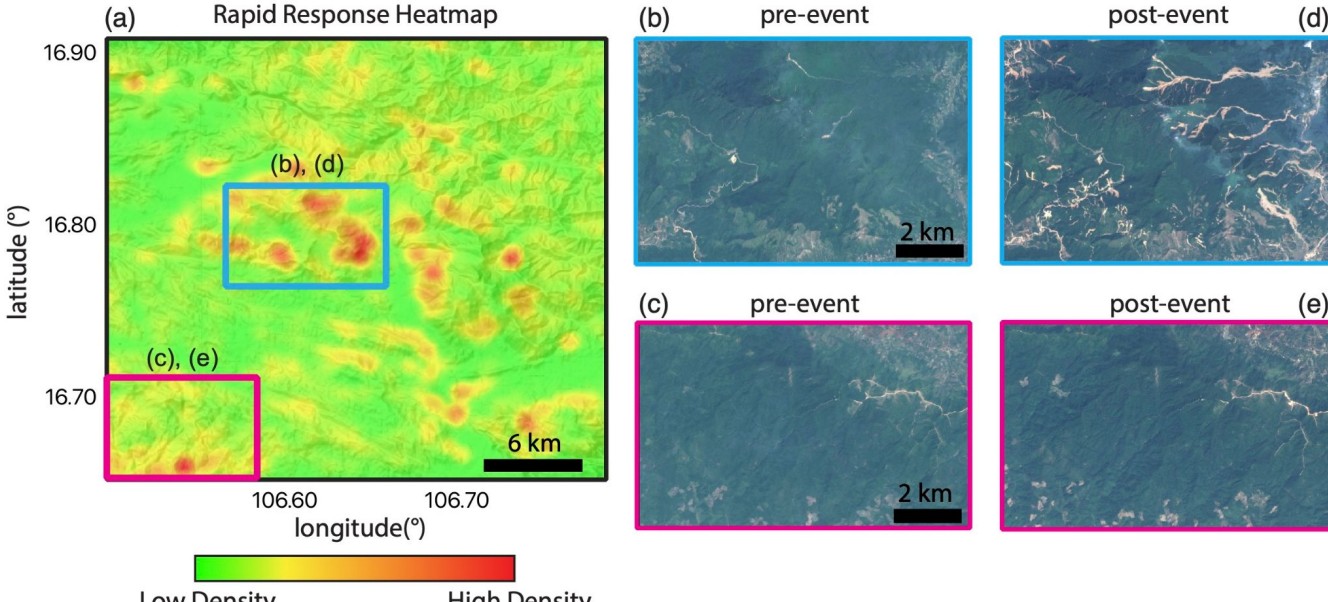

**Figure 7.  Simulated rapid response heatmap for Huong Phung, Vietnam.** (a) Heatmap draped over the NASADEM hillshade of the
topography with red colors corresponding to areas with high potential landslide density. Heatmap made from stacks consisting of 229 pre-
event SAR images collected between 1 October 2016 and 17 October 2020 and 4 post-event SAR images collected between 19 October
2020 and 1 November 2020. Heatmap radius set to 100 pixels or 1 km. DEM mask is used to remove low topographic slope < 10° and
convex curvature > -0.005 m-1. Blue and magenta rectangles show zoomed in areas in figures (b-e). Pre- (b-c) and post-event (d-e) Sentinel-
2 optical images for high and low landslide density zones identified with the heatmap. High density zones in (c) and (e) appear to correspond
to deforestation.

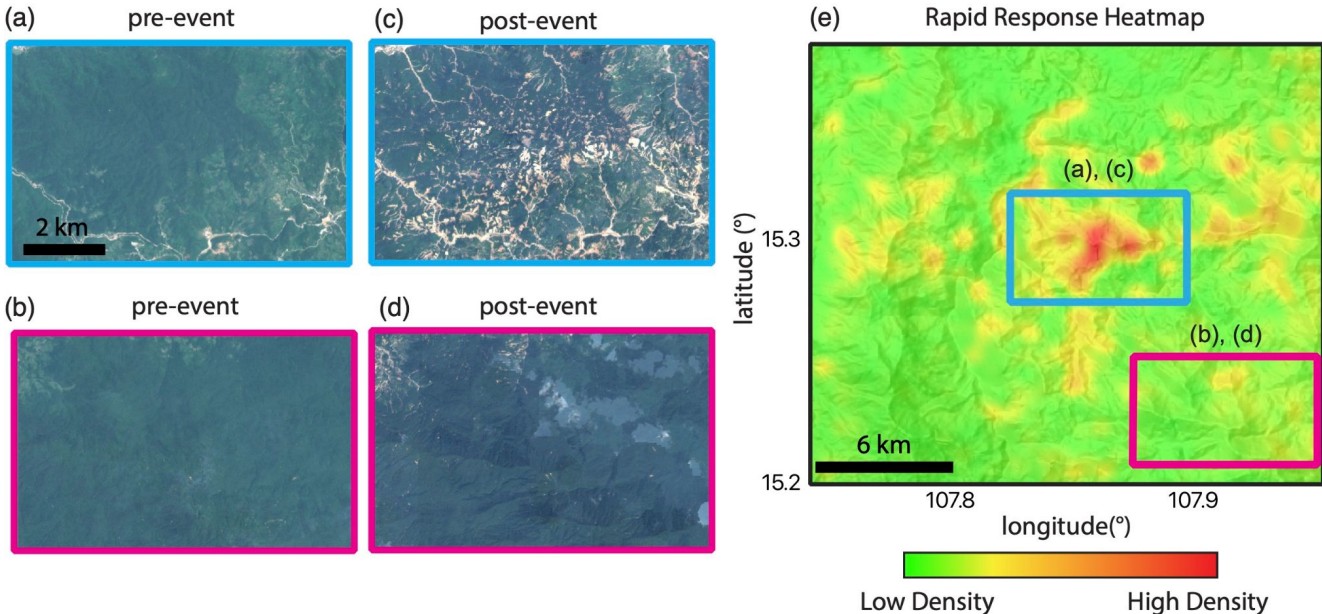

**Figure 8. Simulated rapid response heatmap for Quang Nam, Vietnam.** (a-d) Pre- and post-event Sentinel-2 optical images for high and low landslide density zones identified with the heatmap. (e) Heatmap draped over the NASADEM hillshade of the topography with red colors corresponding to areas with high potential landslide density. Heatmap made from stacks consisting of 253 pre-event SAR images collected between 1 October 2016 and 27 October 2020 and 4 post-event SAR images collected between 29 October 2020 and 7 November 2020. Heatmap radius set to 100 pixels or 1 km. DEM mask is used to remove low topographic slope < 10° and convex curvature > -0.005 m$^{-1}$. Blue and magenta rectangles show zoomed in areas in (a-d).

## 4.5 Application of Landslide Density Heatmaps to Support Rapid Response for 2021 Haiti Earthquake

On 15 August 2021, just one day after the 2021 Nippes, Haiti *Mw* 7.2 earthquake, we used our GEE tools to generate a landslide density heatmap. Our heatmap only includes the landslides triggered by the earthquake and not those triggered by Tropical Storm Grace two days later. We defined the pre-event period between 1 August 2017 and 13 August 2021 and the post-event period between 14-15 August 2021. This resulted in 104 descending pre-event SAR images and 1 descending post-event SAR image. Note because there was only a single descending image collected immediately after the event, we used a descending only stack to create the heatmap. We found an area of high landslide density located ~56 km west of the epicenter (Fig. 9a). We posted our landslide heatmap on the NASA Disasters mapping portal (https://maps.disasters.nasa.gov/arcgis/apps/MinimalGallery/index.html?appid=3b785d8e1ff943e59a9810f67181b8d3) on 17 August 2021, where numerous other remote sensing data sets were hosted including the SAR coherence-based Damage Proxy Map (DPM) and coseismic S1 interferogram that were produced by the Advanced Rapid Imaging and Analysis (ARIA) team at NASA's Jet Propulsion Laboratory and California Institute of Technology in collaboration with the Earth Observatory of Singapore (EOS). The SAR-based products are particularly important because cloud cover has continued to prevent full optical-based landslide mapping for weeks after the earthquake.



The first available optical-based landslide inventory was produced on the day of the earthquake (14 August 2021) using a heavily cloud covered S2 image (Fig. 9b) with the Semi-Automatic Landslide Detection approach (SALaD; Amatya et al., 2021). Due to the cloud cover, only a ~300 km$^2$ area had partial visibility and ~525 of landslides were mapped
(https://maps.disasters.nasa.gov/arcgis/apps/MinimalGallery/index.html?appid=3b785d8e1ff943e59a9810f67181b8d3).   The USGS then released a larger landslide inventory that included 3625 landslides over a ~5000 km$^2$ area between 17-23 August 2021 made from Planet and Maxar optical imagery (Fig. 9a; see data availability). We found good agreement between our SAR-based landslide heatmap and the area of highest landslide density mapped with optical imagery providing further evidence our SAR based approach is well suited for rapid response following major catastrophic landslide events. Additionally,
the initial landslide density map that we produced indicated several other areas of the country that were obscured by clouds and as additional optical imagery became available it was observed that these areas also were impacted to varying degrees by landslide events. A comparison of different landslide datasets and methodologies to support the response and recovery effort is ongoing and will be the topic of future work.

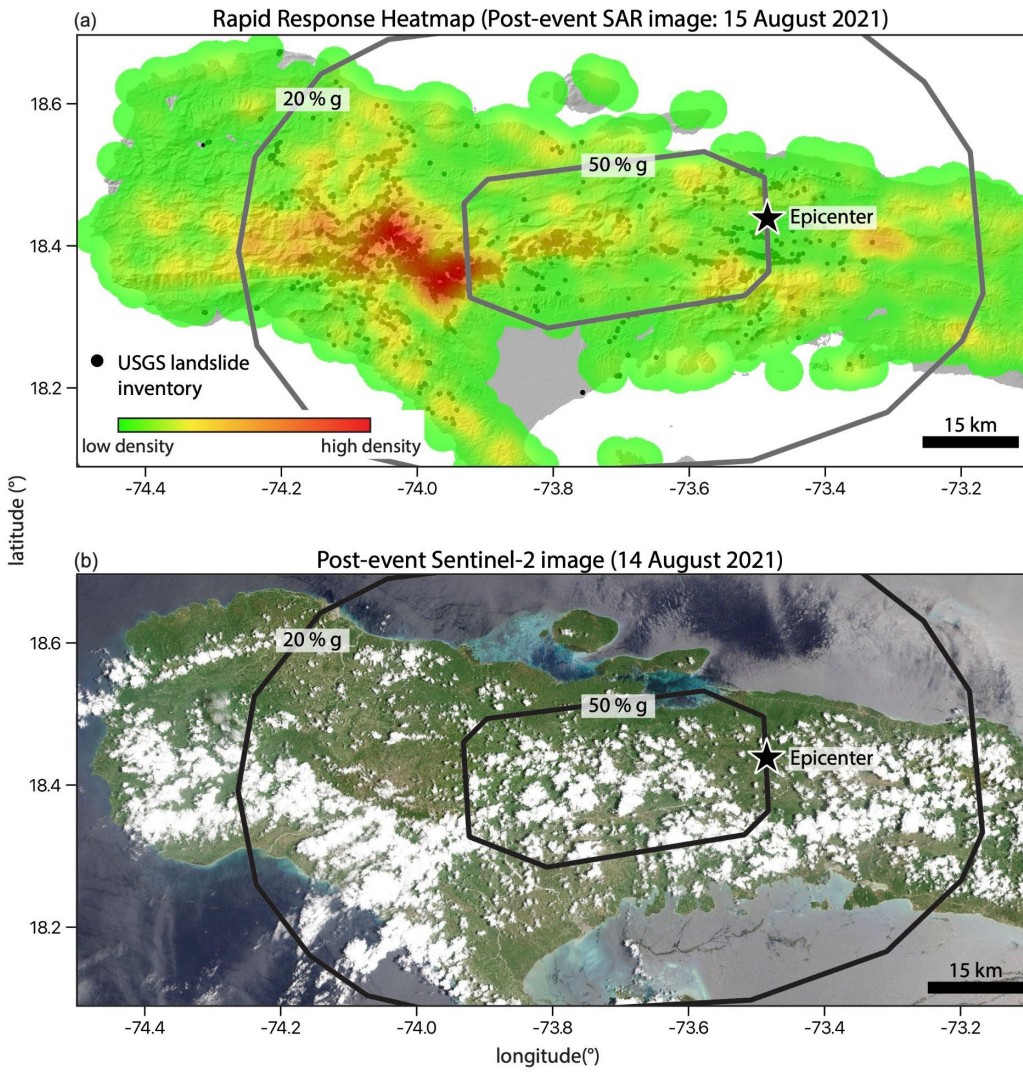

**Figure 9. Real time rapid response heatmap for the 2021 earthquake-triggered landslides in Haiti.** (a) Landslide heatmap draped over the NASADEM hillshade of the topography with red colors corresponding to areas with high potential landslide density. Heatmap made from stacks consisting of 104 descending pre-event SAR images collected between 1 August 2017 and 13 August 2021 and 1 descending post-event SAR images collected on 15 August 2021. Heatmap radius set to 300 pixels or 3 km. DEM mask is used to remove low topographic slope < 10°. Black circles show USGS landslide inventory that was released and updated between 17 - 23 August 2021. The USGS inventory was made from Maxar and Planet imagery. (b) Post-event Sentinel-2 optical imagery showing high cloud cover after on the day of the earthquake. (a,b) Black star shows the $M_w$ 7.2 earthquake epicenter and contours show the peak ground acceleration data as a percentage of gravity (g) provided by the USGS (see data availability).



## 5 Discussion

### 5.1 Landslide Detection using SAR Backscatter Change

Our results show that SAR-based backscatter intensity change in GEE can be used to detect landslides over large areas. The main goal of this study was to develop a methodology for those without SAR expertise or specialty processing software that can be used to create landslide density heatmaps that can aid in rapid response to catastrophic landslide events. We performed sensitivity tests and quantitative analysis for the 2018 Hiroshima landslide event in order to help guide future investigations that do not have an external landslide inventory to help refine their approach. We demonstrated that the landslides in Hiroshima Prefecture, and the other case studies, caused an overall decrease in the SAR backscatter coefficient, which resulted in a relatively large positive $I_{ratio}$ (Figs. 2 and Fig. A1). This decrease in SAR backscatter intensity occurs because the landslide scar and damage acts to decrease backscattering reflectance to the satellite relative to a pre-failure ground surface. Furthermore, we found that the true landslides are well characterized by $I_{ratio}$ values that are $\geq$ 99th percentile (Figs. 4 and 6d).

We found that increasing the total number of SAR images used in the SAR backscatter intensity stacks improved landslide detection performance (Fig. 3). The detection performance was further improved by applying a DEM mask to remove areas where landslides were unlikely to occur. Additionally, combining ascending & descending geometry SAR data into a single stack together improved landslide detection when compared to using ascending or descending data individually (Fig. 3). Combining ascending & descending data into a single stack helps reduce bias introduced from the acquisition geometry (e.g., radar shadows, foreshortening, layover). The combined effect of stacking hundreds of images with both geometries is an improvement from previous SAR backscatter intensity change studies that have focused on individual acquisition geometries and a relatively small number of SAR images. Our findings indicate that future catastrophic events will benefit from a large number of pre-event images (S1 data has been collected since 2014). However, there is likely a point at which there is diminished returns between adding additional pre-event data and computation time. Additionally, we expect further improvements in landslide detection as more advanced SAR processing methods are incorporated into GEE. A recent GEE toolset by Mullissa et al. (2021) implements border noise correction, speckle filtering, and radiometric terrain normalization to the S1 data that may improve our landslide detection capability, but these methods are not yet incorporated into our GEE tools.

The landslide type and size also appear to impact our landslide detection performance. The minimum pixel size of the S1 GRD data is 10 meters, with a pixel resolution of ~3 and 22 m. This resolution limits our ability to detect small landslides with lengths or widths < 20 m and as a result larger landslides are more likely to be detected. Therefore, SAR change detection with S1 data will work better in areas with large landslides, such as the 2018 Hokkaido landslide event. The type of landslide also impacts the detection success and must be considered when setting slope and curvature thresholds. Detection of channelized landslides, such as debris flows, benefits from curvature thresholds that remove convex hillslopes from the analyses, while rockslides, or translational landslides may initially occur along convex hillslopes (e.g., 2018 Hokkaido landslide event). Additionally, removing areas with low slope angles may remove the landslide deposit from the detection





analyses. Therefore, we suggest performing landslide detection with a range of slope and curvature thresholds for each specific field area.

The next step after identifying landslide areas is to construct landslide inventories. The ability to construct accurate landslide inventories generally improves with time when additional post-event SAR images are collected. GEE has drawing tools that can be used to add a marker (i.e., point location), draw a line, polygon, or rectangle. Data from GEE, such as SAR backscatter change maps can also be easily exported as GeoTIFFs for mapping in GIS software. Although not fully explored in this work, we have added a threshold-based approach that classifies pixels with certain $I_{ratio}$ values as landslides and can be used for mapping. This is the same $I_{ratio}$ threshold used to make the heatmaps ($I_{TH,H}$). This approach is included in our GEE

toolset (see code availability). We also note that GEE has Machine Learning capabilities that can be used to help identify landslides.

## 5.2 Landslide Density Maps

Landslide density maps have been used to identify spatial trends in landslide occurrence and to identify areas that were critically damaged during landslide events (e.g, Bessette-Kirton et al., 2019; Burrows et al., 2020; Rosi et al., 2018). These

density maps are also well suited for rapid response because they do not require detailed and time-consuming mapping. Additionally, landslide density maps can also be compared or combined with empirical landslide susceptibility models, which typically operate with a coarse resolution (km-scale pixels), to further refine landslide detection capabilities (Burrows et al., in review).

Landslide density maps are typically made by counting the number of landslides within a fixed area. For example,

Bessette-Kirton et al. (2019) used a 2 x 2 km grid with optical satellite imagery and assigned high landslide density for > 25 landslides, low landslide density for 1–25 landslides, or no landslides for landslides triggered during Hurricane Maria in Puerto Rico, USA. Burrows et al. (2020) used SAR coherence change with S1 and ALOS-2 data at ~200 x 220 m resolution to generate coherence change density maps for landslides triggered by the 2015 Mw 7.8 Gorkha, Nepal earthquake, the 2018 $M_w$ 6.7 Hokkaido, Japan earthquake, and two 2018 $M_w$ 6.8 and 6.9 Lombok, Indonesia earthquakes.

Our landslide density heatmap approach is similar to other density maps, however instead of counting individual landslides we calculate the density of individually detected pixels over a fixed area. That means that the landslide density is calculated at the sub-landslide scale. One major advantage of our heatmap approach with SAR change detection in GEE is that the processing can be done within a short period of time (normally within a few minutes) once the post-event SAR imagery is available in GEE. The results can be easily interpreted and can highlight areas with the highest amount of change detection.

Furthermore, this method can be easily integrated with population density or land use maps, to further prioritize rescue missions after a significant landslide event. For the landslide EOIs we examined, we found our heatmap approach was able to highlight areas with high landslide density for the real time (Haiti) and simulated (Japan and Vietnam) rapid response scenarios.





### 5.3 Challenges with Rapid Response Landslide Detection

Despite the overall success of our case studies, we acknowledge there are many challenges when attempting to detect landslides
for rapid response. In this section, we focus on the challenges and possible sources of error in the rapid response products. The
main challenges of rapid response include: 1) uncertain location and size of AOI, 2) SAR backscatter intensity changes that
do not correspond to true landslides, and 3) computational limitations in GEE.

### 5.3.1 Identifying the AOI and EOI

One major consideration for rapid response is correctly identifying the correct AOI and EOI for the investigation.
This is not a straightforward endeavor and our work here has certainly benefited from performing retrospective analyses with
known EOIs and AOIs (except for Haiti). For rapid response, prompt identification of the location and the scale of an event
soon after a disaster is difficult, and therefore the location and the size of AOI are likely to be somewhat unknown. To help
identify the AOI and EOI for rapid response in real time we recommend consulting the International Charter Space and Major
Disasters (https://disasterscharter.org/) and/or blogs such as The Landslide Blog (https://blogs.agu.org/landslideblog/).
Furthermore, depending on the event, additional information can also be used to identify the AOI. For example, for rainfall-
triggered landslides, satellite-based precipitation estimates, such as those provided by the Global Precipitation Measurement
(GPM) or from the European Centre for Medium-Range Weather Forecasts (ECMWF) can be useful in identifying regions
with higher landslide susceptibility (these products are also available in GEE). There are also freely available global landslide
models that aim to identify likely landslide areas in near real time. Kirschbaum and Stanley (2018) identified potential landslide
activity with the Landslide Hazard Assessment for Situational Awareness (LHASA) model, which combines satellite-based
precipitation estimates from GPM with a global landslide susceptibility map. A second version upgraded LHASA from
categorical to probabilistic outputs (Stanley et al., 2021). LHASA version 2.0 also estimates the potential exposure of
population and roads to landslide disasters (Emberson et al., 2020). These exposure estimates could conceivably be used to
identify the locations where landslides are most likely to have severe impacts. Our heatmap approach could benefit from
guidance and comparison with predictions or "nowcasts" from LHASA. For earthquake-triggered landslides, the users can
incorporate shakemaps and Ground Failure earthquake products produced by the U.S. Geological Survey and set the AOI to
cover the regions with high predicted ground failure or with high peak ground acceleration. We implemented this strategy
successfully for the 2021 Haiti event. Additionally, consideration of population density or township maps can be used to help
determine an AOI.

### 5.3.2 Challenges using the Landslide Heatmap

We found our heatmap approach can be used to identify areas with high landslide density. In order to construct a heatmap, we
had to select an $I_{ratio}$ threshold ($I_{TH,H}$), kernel density shape, and kernel density radius. We found that a $I_{TH,H} \geq$ the 99th percentile
of the entire AOI characterized the true landslides in all case studies. However, because the $I_{TH,H}$ is based on the 99th percentile




value of the entire AOI, by definition the heatmap will always highlight some pixels (e.g., false positives) even if there is no
landslide event. In many cases, these false positives will be related to natural and manmade ground surface change, including
but not limited to deformation from mining, construction, deforestation, agriculture, flooding, snow cover, and changes in
reservoir water levels. We observed numerous places where SAR backscatter coefficient change is not related to landslides
(Figs. 2c and 2d, 4, 6c and 6d). The DEM mask helps remove some, but not all of these false positives. Land use/cover maps
have also been successfully implemented to help reduce false positives (e.g., Adriano et al., 2020), but have not been applied
in our work. If there is no actual ground deformation in the AOI, we expect the 99th percentile pixels would be randomly
distributed, and therefore the heatmap will not highlight a specific region.

To further explore potential issues with our heatmap approach we constructed a heatmap for our Hokkaido AOI
during a period of time with no known landslide events just prior to the Iburi earthquake (Fig. 10). We set the pre-event time
period from 1 August 2015 to 5 June 2018 and the post-event time period from 6-21 June 2018. Our heatmap detected a large
area near the center of the AOI. Upon further inspection, we found this change in backscatter intensity was related to filling of
a large reservoir. The reservoir was not entirely masked out of our heatmap because the edges of the reservoir area have slopes
> 10°. While we expect false positives to exist using this method, we show that overall, the high density of ground surface
change after a known landslide-triggering event points to the overarching spatial distribution of landslide activity when slope
and other filtering is applied.

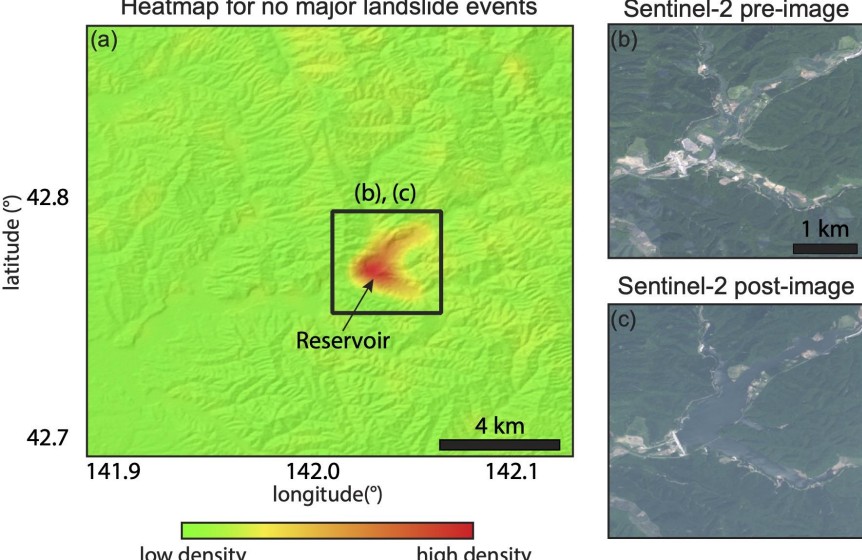

**Fig 10. Example heatmap made during a period with no major landslide events in Hokkaido, Japan.** (a) Heatmap draped over the NASADEM hillshade with red colors corresponding to areas with high density of ground surface change. Heatmap made from stacks consisting of 127 pre-event SAR images collected between 1 August 2015 and 6 June 2018 and 5 post-event SAR images collected between 6 - 21 June 2018. Heatmap radius set to 100 pixels or 1 km. High density region in heatmap corresponds to true ground surface change related to infilling of a large reservoir. Because the slopes of the reservoir have slopes > 10°, they were not masked out of our analyses. (b,c) Pre- and post-image Sentinel-2 optical images showing the infilling of the reservoir.



### 5.3.3 Computation Limits in GEE

Although GEE enables analyses of large quantities of data, it is a shared cloud computing resource and as such it has user
limits. GEE restricts the total number of simultaneous processing requests and the maximum duration of requests and computational memory (Gorelick et al., 2017). Thus, computational issues can arise in GEE when trying to process extremely large datasets (> 500 images) over large areas (> 3500 km$^2$). To overcome these issues, we suggest starting with small AOIs and then enlarging the AOI or using multiple AOIs instead of one large single AOI. Additionally, for events that occur after 2021, there is a large quantity of pre-event data that may be redundant and only increase computational time. Thus, we suggest
limiting the pre-event time period to < 4 years.

### 5.4 Satellite Acquisition Frequency and Landslide Detection

Given the relationship established in Section 4.1 between the number of images used in our pre- and post-event stacks and AUC (Fig. 3), we explored how the satellite revisit frequency, and hypothetical changes in revisit frequency, could impact landslide detection. To better understand the relationship between number of images and landslide detection success, we used
all of the available pre-event images and continued to increase the $T_{post}$ duration following the rainfall event to determine when we would achieve the maximum AUC for the Hiroshima case study (Fig. 11). The goal of this exercise was to determine how many post-event SAR images are needed for the best landslide identification shown in Section 4.1. This information is critical for understanding the level of potential SAR detection success to expect for future landslide events when no external landslide inventory exists. We found that the AUC continues to increase with post-event time, and thus the number of images acquired.
The AUC is > 0.7 in just 28 days (6 post-event SAR images) after the event and reaches a maximum of 0.8342 at 684 days (136 post-event images) after the event. This increase in AUC can be approximated as an increasing form of exponential decay, with a rapid increase in AUC shortly after the event followed by a slower increase in AUC several months after the event (Fig. 11a). By plotting the AUC scores with time, we found that the AUC scores increase roughly linearly with log time (Fig.11b). Finally, using our AUC score model fits, we simulated the AUC score for hypothetical changes in the satellite revisit time.
Our findings suggest that if the satellite revisit was twice the current revisit time, the modeled AUC score would be ~0.7 just 1 week after the EOI, while if the revisit time was half the current revisit the modeled AUC score would be ~0.65 (red lines in Fig. 11b).

Although we found that AUC score increases following the EOI, the rate of AUC increase is temporarily reduced between 56 and 183 days after the event (Fig. 11a). We infer that this change in the rate of AUC increase is due to seasonal
changes in vegetation. The image stacks made between 56 and 183 days after the event contain a relatively high number of images collected in fall and winter (between November 2018 and February 2019) when vegetation cover is likely reduced relative to the average yearly vegetation cover represented in the long-term pre-event stack. The change in the rate of AUC increase is clearly shown by examining the fitted lines in Figure 11a. The rate of the AUC score increase returns back to the overall trend after the spring season, possibly due to the growth of vegetation. Seasonal changes in land cover must be taken




into account when using SAR intensity change to identify landslides as these seasonal changes can result in false positives or obscure true positives (Jung and Yun, 2020). We are able to overcome seasonal changes by using a large quantity of data to effectively smooth over seasonality, but different strategies may be employed in cases where data is limited (i.e., examine data all from the same season).

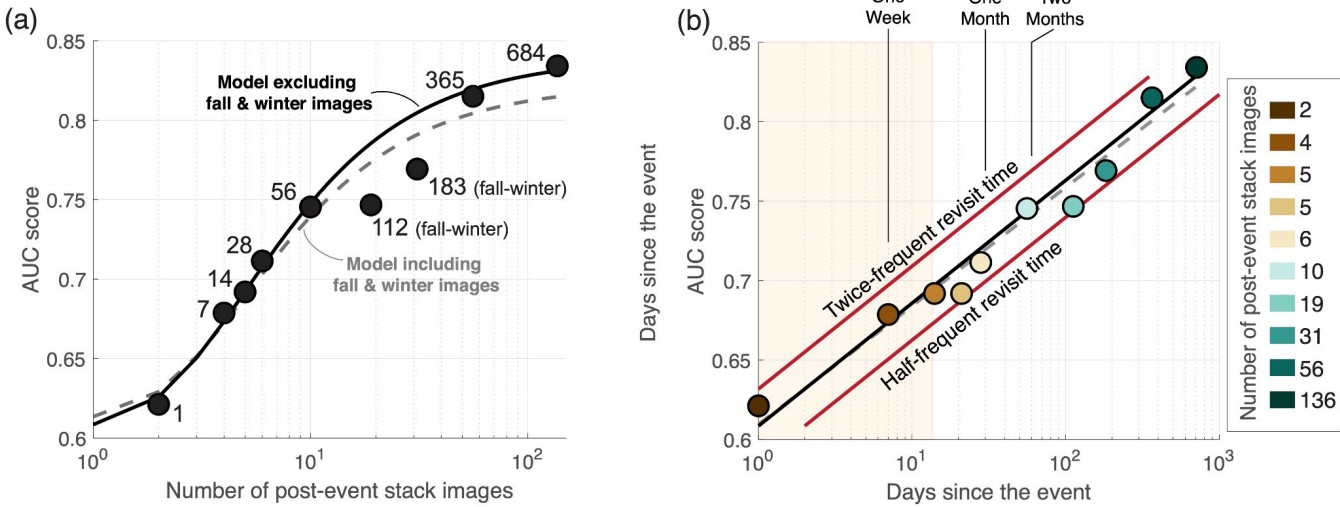

**Figure 11.  Area Under the Curve (AUC) score as a function of increasing number of days and SAR images following the Hiroshima landslide event.** (a) AUC score as a function of the number of SAR images used in the post-event stack. Black dashed and solid lines show exponential fit models with and without fall-winter SAR images, respectively. The rate of AUC increase decays exponentially with the number of images in the stack. Numbers next to black circles correspond to the number of days since the landslide event. The two fall-winter SAR images labeled fall off the best-fit line, which we infer is due to seasonal differences in SAR backscatter intensity. (b) AUC score as a

function of days since the Hiroshima landslide triggering event. The dashed line shows the AUC score fit model and the black line shows a model fit excluding fall-winter images. Light orange rectangle highlights the two week long rapid response time period. The red lines indicate the predicted AUC score time series by assuming a satellite revisit time that is twice or half the S1 satellites for this region. Color scale corresponds to the number of images used in the post-event stack.

**5.5 Future Work**

We only explored SAR backscatter intensity change with the C-band Sentinel-1 satellites because these are the only SAR data freely available in GEE. However, our methodology can also be applied to SAR satellites operating with different radar wavelengths and different pixel resolutions (e.g., X- and L-bands). Recent work by Adriano et al. (2020) used SAR intensity change detection with data from the JAXA ALOS-2 satellite to identify landslides for the same EOI in the Hiroshima Prefecture. ALOS-2 PALSAR-2 data has a L-band radar (~24 cm radar wavelength), which better penetrates through dense

vegetation. They were able to clearly detect many landslides in their study area. Unfortunately, we were not able to make a direct comparison with their dataset due to data availability. Similarly, Jung and Yun (2020) and Burrows et al. (2020) also used ALOS-2 data with SAR intensity change, as well as coherence change methods to detect landslides after the 2018 Hokkaido landslide event. Jung and Yun (2020) found that a multitemporal intensity correlation method provided the best landslide detection in the vegetated mountains of Japan. Unfortunately, ALOS-2 is collected relatively infrequently worldwide

and is not freely available, which limits the use of our multitemporal and open-source SAR backscatter change approach with



these data. The NASA-ISRO SAR (NISAR) mission, which is currently expected to launch in January 2023, will operate with an L-band (~24 cm) SAR sensor and is designed to fly by the same location every 12 days. As L-band data can generally produce improved results in vegetated regions (Yun et al., 2015; Jung and Yun, 2020; Burrows et al., 2020), we expect an improvement in our multi-temporal stacking SAR backscatter change approach to detecting natural hazards using images from
the NISAR mission. Similar to the Sentinel program the NISAR products will be publicly available. If GEE also ingests the NISAR products, the same GEE scripts provided in this study can be used for the NISAR images.

Although we were able to successfully identify many landslides, we expect that SAR backscatter intensity change for landslide detection may require different processing strategies in different environments. For example, landslides that occur in regions with different types of land cover, or in regions that have significant seasonal changes (e.g., snowfall, vegetation cover)
may require all pre-event and post-event data to be from the same season. Also, SAR data collection is not the same in all places around the world. For areas that have more frequent S1 data collection, we expect better ability to rapidly identify landslides, while the opposite is true for regions with less data collection. For our future work, we will use our GEE approach to explore how multi-temporal backscatter intensity change stacking identification methods performs in different environments and in different climates given each will have different amounts of available S1 data. We will also test if the recently released
GEE package for border noise correction, speckle filtering, and radiometric terrain normalization can improve our landslide detection capability (Mullissa et al., 2021). Nonetheless, our methodology presented throughout this manuscript, including stacking strategies and $I_{ratio}$ values provide a good starting point for all landslide events worldwide.

## 6 Conclusions

In this manuscript, we developed a new methodology to detect landslides (and other ground surface changes) using freely
available SAR data, topographic data, and open source tools in the Google Earth Engine platform. Our approach does not require specialized SAR processing software, and furthermore, it does not require the user to download large volumes of data to a local system. We found that the ratio of two multi-temporal SAR backscatter intensity image stacks, composed of pre- and post-landslide event data, can detect areas with high landslide density for rapid response (within two weeks of a landslide event) and event inventory analyses. We found the best strategy to detect landslides was to combine all available SAR images
acquired on ascending and descending satellite flight paths with topographic data to mask out areas that were unlikely to experience landsliding and to construct landslide density heatmaps. Importantly, we found that landslide detection capability increases rapidly over the first two months and then continues to increase slowly with more image acquisitions. This finding implies that satellites with higher repeat acquisitions may provide more accurate landslide identification that can assist with rapid response. Alternatively, SAR data operating with longer radar wavelengths will help reduce noise and could improve
landslide detection, especially for rapid response. Future SAR missions, like the L-band NASA-ISRO NISAR mission, which is currently expected to launch in January 2023, will also provide publicly available data. If Google Earth Engine ingests the





NISAR data, our methodology could be used for Sentinel-1 and NISAR, which will undoubtedly improve the ability to detect and monitor natural hazards.

**Data and Code availability.** The data used in this manuscript were provided by the National Aeronautics and Space Administration (NASA) and the European Space Agency (ESA) Copernicus program and accessed on Google Earth Engine (https://code.earthengine.google.com). The Geospatial Information Authority of Japan (GSI) and Association of Japanese Geographers (AJG) 2018 Hiroshima landslide inventory is available at https://ajg-disaster.blogspot.com/2018/07/3077.html. The 2018 Hokkaido landslide inventory from Zhang et al. (2018) is available at

https://zenodo.org/record/2577300#.YRCONu1lDUJ. The 2021 Haiti landslide inventory from the USGS is available at https://usgs.maps.arcgis.com/home/item.html?id=bbd6cb29e5a64ed380e7ee14e820d673. Other landslide and remote sensing data for the Haiti earthquake are available at https://maps.disasters.nasa.gov/arcgis/apps/MinimalGallery/index.html?appid=3b785d8e1ff943e59a9810f67181b8d3. Additional Haiti earthquake data available at https://earthquake.usgs.gov/earthquakes/eventpage/us6000f65h/executive.

Google Earth Engine Code for landslide detection on the Google Earth Engine is available at https://github.com/MongHanHuang/Codes-for-Handwerger-et-al-2021-preprint.

Codes used to generate figures are available through the following Google Earth Engine links:

Figure 2a (https://code.earthengine.google.com/5a33f029c505bbc84c182703e4a8a37e),

Figure 2b (https://code.earthengine.google.com/9c495a90854df33d454aa3c2a42b11db),

Figure 2c (https://code.earthengine.google.com/9bcea4853c1b3d588aa321fc05a00c8c),

Figure 2d (https://code.earthengine.google.com/6fa0ffbf23e07920fa5f4f4bf030ed05),

Figure 4a (https://code.earthengine.google.com/2a6616eb4b6adcaea717fa9e4db95eaa),

Figure 4b (https://code.earthengine.google.com/5cb2479ac02cf97da81d228f302b7b9b),

Figure 4c (https://code.earthengine.google.com/cdf57aec21b81fbac55ae1bd0cb20249),

Figure 4d (https://code.earthengine.google.com/a75a9576aba88f3295186ac8fc27ba40),

Figure 4e (https://code.earthengine.google.com/c806f5c1eec3bc74ad1f27e2e1ebfa8f),

Figure 4f (https://code.earthengine.google.com/eaba5dd178dd0f0233e80a7f3f7b6230),

Figure 5 (https://code.earthengine.google.com/30b4fcf62327afaf5d399e81cc671c75),

Figure 6a (https://code.earthengine.google.com/fc7c16d92aee9c3817615ff4450f2958),

Figure 6b (https://code.earthengine.google.com/59fe2cdbd3aced46ef5349b23666ebfc),

Figure 6c (https://code.earthengine.google.com/c7c26cba6fa0f314241ba64420082d1d),

Figure 6d (https://code.earthengine.google.com/059e0d1e3ea2f29c8fe2f3bc90d2a59b),

Figure 7a (https://code.earthengine.google.com/4d82542bf29bcc5d62b4f3ee4f231e73),

Figure 7b (https://code.earthengine.google.com/374a9ebc926ffe6b19e430f16a8bb43d),

Figure 7c (https://code.earthengine.google.com/d4aaa87b4bc4bdb62a17589fac2287d6),





Figure 7d (https://code.earthengine.google.com/26078cfefc4e53b82607213fdca427cd),

Figure 7e (https://code.earthengine.google.com/522f8a7abfeab866560f215ecab1cec6),

Figure 8a (https://code.earthengine.google.com/2f35694a5f1137fbd0661e1874a88581),

Figure 8b (https://code.earthengine.google.com/a8140ba088b062b8664b10b2b00950ab),

Figure 8c (https://code.earthengine.google.com/956c5b64b3fd163f6e07005237ee4203),

Figure 8d (https://code.earthengine.google.com/6b2c091fb415d249cbff4c9512dda280),

Figure 8e (https://code.earthengine.google.com/8431794865d2c8c41043c145ba6fefe0),

Figure 9a (https://code.earthengine.google.com/a09153b99eb5c2290c01ea316c92aa18),

Figure 9b (https://code.earthengine.google.com/f9763df2946f1b690ae08094134b1f65),

Figure 10a (https://code.earthengine.google.com/245f1e90270ed17da8cdd0324fbeec1f),

Figure 10b (https://code.earthengine.google.com/1da44c28745901065545acafc6f87cf2),

Figure 10c (https://code.earthengine.google.com/3c0acb9fd8ec891c825f2b27abc07f91).

**Author contribution**. ALH, SYJ, PA, and MH designed the study and processed and analyzed the data. MH and ALH wrote

the Google Earth Engine codes. ALH and MH performed the geomorphic analysis. All authors performed landslide interpretation. PA and DBK identified the test sites and provided advice and guidance on landslide detection. MH and HRK performed ROC analyses. ALH and MH wrote the manuscript with contributions from SYJ, PA, HRK, and DBK.

**Competing interests**. The authors declare no competing interests.


**Financial support.** Funding for this work came from NSF PREEVENTS-2023112 grant (ALH), NSF EAR-2026099 (MH), and High Mountain Asia NNX16AT79G and Disaster Risk Reduction and Response 18-DISASTER18-0022 (DBK and PA).

**Acknowledgements**. We thank the National Aeronautics and Space Administration (NASA), the European Space Agency

(ESA) Copernicus program, and the Google Earth Engine for providing freely available data and processing. We thank the Geospatial Information Authority of Japan (GSI) and Association of Japanese Geographers (AJG) for providing the Hiroshima landslide inventory, Zhang et al. (2018) for providing the Hokkaido landslide inventory, and the USGS for providing the Haiti landslide inventory. Part of this research was carried out at the Jet Propulsion Laboratory, California Institute of Technology, under a contract with the National Aeronautics and Space Administration (80NM0018D0004).





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

**Appendix A: Additional figures.**

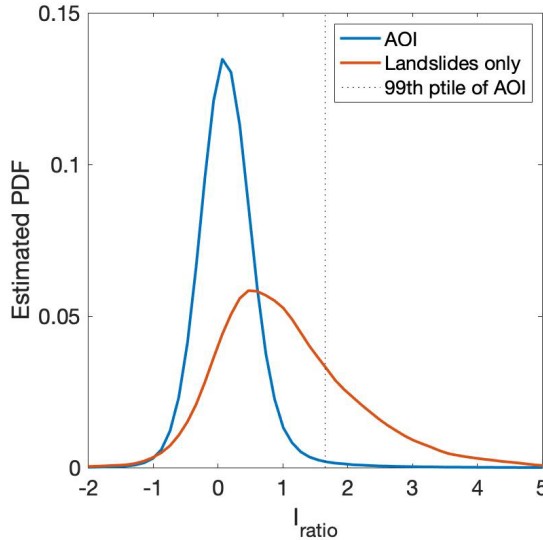

**Figure A1: Kernel density function of $I_{ratio}$ values for the 2018 Hiroshima landslide event.** $I_{ratio}$ distribution for the full AOI including both landslide and non-landslide areas. (b) $I_{ratio}$ distribution for the true landslides mapped by GSI/AJG. $I_{ratio}$ calculated using combined ascending & descending stacks from 142 pre-event SAR images collected between 01 May 2015 and 29 June 2018 and 133 post-event SAR images collected between 09 July 2018 and 29 May 2020. This plot corresponds to our "best case" landslide detection described in section 4.1.
