# Peer review of "Strategies for landslide detection using open-access synthetic aperture radar backscatter change in Google Earth Engine"

_Natural Hazards and Earth System Sciences, 2021_

## Author Comment (AC1)

**The author's response is shown below in blue text.**

**Anonymous Referee #1**

I do like the revised version of the paper. The novelty and usefulness of the described approach is more clear now. I did not see that so clearly in the previous version, which might be a) my fault as being ignorant or b) the new version is a significant improvemtn or c) both. I do like the heatmaps and I think they are useful as is the new experiment in Haiti.

We thank Referee #1 for their review of our manuscript. We are glad to hear our effort to revise the manuscript based on previous reviews has led to an improved contribution.

I only have a few minor comments:

Line 48: Delete 'freely available' because the statement you make hear does not depend on the free availability of the data.

Good point. We will delete this from the revised manuscript.

Line 60: "...in densely vegetated mountainous areas where landslides tend to occur." I don't think this statement is correct, as many landslides also occur in not densely vegetated areas. I even believe that vegetated areas have potentially less landslides.

Another good point and we agree. We will remove the statement about vegetation and modify the sentence to "However, backscatter change detection methods can outperform coherence-based methods in densely vegetated mountainous regions, because in these areas…"

Line 119: "...the dominant scattering is away from the satellite..." - I suggest to delete this statement. It is unnecessary here and probably not true in all cases -> dominant scattering might be back to the satellite but very weak or, even more likely, there is no dominant scattering direction.

Agree. We will delete this from the revised manuscript.

Line 271: I think EOI was not introduced before

EOI (event of interest) is introduced on Line 135.

---

## Author Comment (AC2)

**The author's response is shown below in blue text.**

**Anonymous Referee #2**

The authors propose a method and some best practices / strategies for landslide detection and/or landslide density headmaps estimation using multitemporal SAR images pre- processed in google Earth Engine. The method combines quite traditional approaches including change detection thresholding and geomorphological filters.

The landslide density headmap based workflow is tested for five different events.

Despite quite clear writing, the manuscript is not, in my opinion, very readable. This is probably a consequence of a research framework with some ambiguities and inconsistencies, and the use of a bit too rough of some concepts. This makes it difficult to say whether the conclusions are really supported by evidence.

In the end, I can't find in this work any substantial scientific significance and it seems to me too hasty to assume that it contributes to defining new strategies or best practices for landslide detection.

We thank the Referee #2 for their honest opinion and their review of our manuscript. Below we will respond to their concerns individually.

My main concerns are referred to:

1) an unclear (probably inconsistent) use of the Hiroshima test to define best practices based on algorithms and products (ROIs, and detection) that are not used in the following test cases. There is no evidence that those best practices are methods invariant.

In our paper we use the Hiroshima test case to determine best practices for landslide detection, and we use these same approaches to successfully detect landslides at 4 other test sites. This includes using large pre-event data stacks, combined data from ascending and descending tracks, and a DEM mask to remove areas unlikely to experience landslides. Furthermore, we do not claim that our best strategies are method invariant. In fact, we discuss a variety of scenarios where our strategies may need to be adjusted in Section 5.3.

2) a lack of quantitative validation when the threshold is chosen manually (nothing against expert-driven methods, but they have to be supported by evidence and validation. Any consideration on FN is missing.

We provide quantitative validation for the Hiroshima test case. We do not provide quantitative validation of the other 4 test sites because the goal is to detect areas with high landslide density for rapid response when no pre-existing landslide inventory is available. Furthermore, our qualitative validation shows that we are able to successfully detect landslides at all test sites. In this way we adapt the methodological framework used for several recent NHESS publications, such as Burrows et al. (2020) and Scheip and Wegmann (2021) (i.e., ROC analyses and direct

visual comparison with optical imagery). Nonetheless, we will provide a new figure in the Appendix (also shown below in this document) that will allow the reader to see how changes in thresholds impact the landslide detection.

We are not sure what FN means here, but will guess that it means False Negatives? Referee #2 is correct that we did not consider FNs because this requires direct comparison with individual landslide polygons. Instead, we chose to develop a landslide density approach, which does not focus on individual landslide polygons, but areas of high landslide density. To better explain this point, we will add text to the revised manuscript that states "We are not able to directly assess the number of false negatives with the heatmap because this approach is not used to map individual landslide polygons"

3) some decisions taken on an apparently superficial knowledge of the data and algorithms used (see some comments later)

the three points are addressed in my following comments to the manuscript.

I'm sorry to suggest rejecting the paper but I encourage the authors to go through a more correct definition of the purposes and of the research framework.

###############################################################
#################################################

Introduction

89 experienced landslide activity: correct, but I'd like to suggest not to use activity here because it can be confused with the activity as defined in https://doi.org/10.1007/s10346-012-0335-7 (same at line 95)

We agree this is confusing and we will delete the word "activity" in the revised manuscript.

2 Methods
2.1 SAR backscatter in Google Earth Engine

126 For this study we only used SAR data in the VH polarization: in a number of papers cited in the manuscript also VV is used with a few advantages compared to the depolarised signal in particular when roughness does not matter much. I think the choice of providing best practices just using a single channel should be better introduced.

While we agree with you that VV data are also useful, we are not able to test these data in our manuscript. We have made it clear that we only used VH data, therefore there should be no confusion regarding comparison with VV data. Furthermore, we specifically encourage other users to explore VV data, when possible, and we include citations to other studies that use VV data.

2.2 Landslide Detection Approach

132 The AOI can be a single landslide or a mountain range: it sounds a bit weird because it seems to assume that the landslide is already known. Also, the density map seems to me to lose/change meaning

We agree with the Referee's point and we will change the sentence to read "The AOI can be a small region or a large mountain range".

135 full event inventories: which does not seem the real purpose of the procedure (density)

Our tools can be used for rapid response and making full event inventories. For rapid response, the heatmap approach is suggested. For event inventory mapping the suggestion is to use many post-event images to analyze the $I_{ratio}$ data (e.g,. Figure 2c). We also documented the ability to detect landslides for event inventories with ROC analyses (Figure 3a,b). While event inventory mapping is not the focus of our work, the same approaches can be used to construct inventories, especially in regions with persistent cloud cover, and therefore we think it is important to state this information.

137 - 142 We also ... stacks: this part needs some more clarifications, in fact, stacking per se does not reduce noise. I guess the authors assume that the mean of the images should be less noisy and more representative of the ideal surface backscatter of the different land covers in the area. Since here we are dealing with random processes, this should be true/acceptable when the series is stationary. Is this true in all the test areas? I suggest to explicit when this is not applicable. furthermore, the atmospheric delay is not canceled by averaging the pixel values but it is smashed in the series.

Referee #2 is correct that in stacking we find that the SAR data become less noisy and more representative of the pre- or post-event ground surface backscatter. We also understand the Referee's point about atmospheric delay. In the revised manuscript we will clarify this point by modifying the text to read "We also reduce transient noise and error by stacking images to create pre-event ($I_{pre}$) and post-event ($I_{post}$) backscatter intensity stacks. SAR data stacking has been shown to significantly improve the signal-to-noise ratio in SAR data (e.g., Cavalié et al., 2008; Zebker et al., 1997). Thus, the SAR image stacks provide backscatter data that is more representative of the pre- or post-event ground surface properties."

Our results show stacking works well for our 5 test sites and this is our recommended method. We did, however, describe in the manuscript that different strategies may be needed for different land covers or in different environments in the Discussion Section. Our manuscript states "we expect that SAR backscatter intensity change for landslide detection may require different strategies in different environments. For example, landslides that occur in regions with different types of land cover, or in regions that have significant seasonal changes (e.g., snowfall, vegetation cover) may require all pre-event and post-event data to be from the same season. For our future work, we will use our GEE approach to explore how multi-temporal backscatter intensity change stacking identification methods perform in different environments and in different climates given each will have different amounts of available S1 data."

155 -159 Since ... deposits. It is very/too generic. I suggest linking this to the very wide literature on the topic (i.e. susceptibility)

Our main point here is that landslides sometimes runout and deposit onto low slope areas, which may be masked out if using a slope-based threshold mask.

2.3 Change detection performance and determination of most effective detection strategies

164 – 165 quantitatively evaluated our results with a previously published landslide inventory for the 2018 Hiroshima landslide event using Receiver Operating Characteristic curves (ROC) (Fan et al., 2006): It seems to me that this is not evaluation but calibration: usually, thresholds are found in a part of the study area, and then applied to the remaining part of the area for independent evaluation. Here a step is missing.

We thank Referee #2 for their comment; however the ROC analyses are not a calibration step. The ROC curves are calculated by exploring all possible $I_{ratio}$ thresholds. i.e., we do not pick a threshold based on the ROC. We then quantify the landslide detection success using the AUC (area under curve). In other words, the ROC is only used as a metric of how well the SAR change data captures the true landslides.

171 – 174 We quantified ... pixels: see my previous comment: also this sentence can sound ambiguous. ROCs are used here to find the best threshold which makes the benchmark and the new product as 'closest as possible.

This is incorrect. The ROC is not used to pick a threshold, it is used to calculate the detection success.

178 10 m x 10 m: this is the pixel size and not the spatial resolution (should be something like 20 m but it really depends on the filtering), so is 100m2 enough? As far as I can understand, images were not filtered, so, dealing with single/very few pixels is quite risky (speckling, multiplicative noise, and so on...)

We thank Referee #2 for their comment. We believe that 100 $m^2$ is sufficient for comparison to the previously published Hiroshima inventory (which contained many landslides <100 $m^2$ ), but acknowledge that the S1 data is better suited at identifying larger landslides. Removing landslides < 400 $m^2$ (i.e., 20 m x 20 m) would only serve to improve our AUC score as the larger landslides are better detected. We will also add a new sentence to the revised manuscript that states "Furthermore, users should exercise caution interpreting individual pixels that are isolated in space as landslides. Landslides are most often comprised by clusters of nearby pixels"

204 – 205 We manually explored I TR,H using I ratio percentiles to find the threshold value that visually highlights true landslides and reduces noise: in my opinion, a qualitative analysis here is not enough, furthermore a single test case is too dependent on the event. More events should be compared a priori (before the density map)

We thank Referee #2 for their feedback. We will add a new figure to the Appendix (new Figure A1) to show how the landslide detection changes for 99th, 90th, and 80th percentile for the Hokkaido event and two Vietnam events. It is clear that while the number of true positives increases, the number of false positives increases even more, which will bias the result of the heatmap. The 99th percentile corresponds almost entirely to true positives. We cannot perform a quantitative analysis at all sites because this requires us to compare to a pre-existing landslide inventory (which does not exist for Vietnam or Haiti).

[Figure]

**Figure A1: Detected landslides based on $I_{ratio}$ threshold values ($I_{TR,H}$).** Sentinel-2 (S2) post-event optical imagery and 99th, 90th, and 80th percentile- thresholds for detecting landslides for (a-d) Hokkaido, Japan, (e-f) Huong Phung, Vietnam, and (i-l) Quang Nam, Vietnam.

208 -209 must be defined without the use of an external landslide inventory: then with what? By using an optical image? But this is against the starting presuppositions. furthermore, this invalidates the fact that the best practices found with ROC can be directly used here because the approach used in an intermediate step is different...

We apologize for the confusion here. This is exactly where the percentile-based threshold is used. We found that the 99[th] percentile works as a threshold for all test sites and so the threshold is automatically calculated in the GEE codes by finding the 99[th] percentile.

3 Test sites

3.1 2018 Rainfall-triggered Landslides, Hiroshima, Japan

3.2 2018 Earthquake-triggered Landslides, Hokkaido, Japan

3.3 2020 Rainfall-triggered landslides, Huong Phung and Quang Nam, Vietnam

3.4 2021 Earthquake-triggered landslides in Haiti

4 Results
4.1 Determining Effective Strategies for Detecting Landslide

The use of the Hiroshima event and ROIs to define the most effective strategies is really confusing me, in fact, the results are dependent on the approach used and the final product which here are ROIs and pixels potentially inside landslides, while in the main part of the paper other strategies to find thresholds are used, and the final product itself is different: density. How can you make sure that what is eventually true for one method and one type of product are still true when other methods are applied, and furthermore to obtain different products?

We apologize for the confusion. The main point of our paper is to show that we can apply our best strategies for Hiroshima to 4 other test cases that are in different environments and that our heatmap approach works well to visualize areas with landslide density. Thus we are using the pixels defined as landslides to create a visualization that highlights areas with many landslides. We have text in the manuscript that explains how some of the methodology may need to change in Section 5.3.2. We further note that this study is not meant to test all possible products with all possible methods. Our main goal is to document a method that uses open-access tools and data to detect areas with high landslide density for rapid response.

311 -312 (i.e., all available pre-event
and post-event data) and found the slope and curvature thresholds that maximized the AUC: since this is event dependent, how much this result can be generalised?

Slope and curvature thresholds for landslides can be somewhat generalized (for instance, landslides occur on slopes > 12 deg; Adriano et al., 2020), but we mention in the manuscript that these thresholds may vary from place to place and with landslide type. It is always important to incorporate local knowledge of landslides and topography when possible, but general slope/curvature thresholds can be used as a starting point.

326 – 328 lastly ... values: as far as I know, the ratio (of two gammas) should be beta-1 and the log of beta-1 is not in the list of the known pdf. I suggest not to include this analysis or to delve into more.

To simplify, we will follow the Referee #2's suggestion and remove these lines from the revised manuscript.

4.2 Determining Effective Strategies for Rapid Response 338 making comparison: it is not a comparison but a tuning

As mentioned above there is no tuning involved in ROC analyses. All possible thresholds are used to construct the ROC, the AUC is a measure of true positives to false positives.

344 -345 low signal to noise ratio of the post-event images: due to problems in the acquisition? Processing?

Here we are explaining that there is a lower signal-to-noise ratio compared to the post-event stack when more post-event data are acquired. We would need external ground based data to determine the source of the noise.

4.3 Landslide Detection with SAR-based Change Landslide Density Heatmaps

352 - 353 the true landslides and removes most of the false positives: this is to me a weak point of the work, a quantitative analysis should be carried out, showing for example also false negatives.

We thank Referee #2 for their comment. We will add a new figure to the Appendix (also included in this response document above) of the revised manuscript to show how the percentile-based $I_{ratio}$ detects landslides for other test sites. Also mentioned above, we cannot directly assess false negatives because this requires a direct comparison with individual landslide polygons. We will add the following to the revised manuscript "We are not able to directly assess the number of false negatives with the heatmap because this approach is not used to map individual landslide polygons. "

5 Discussion
5.1 Landslide Detection using SAR Backscatter Change

477 -478 This decrease in SAR backscatter intensity occurs because the landslide scar and damage act to decrease backscattering reflectance to the satellite relative to a pre-failure ground surface: the sentence is messed up, just say that landslide cause a decrease (in this particular case) of the radar backscatter..

To clarify, we will delete this sentence from the revised manuscript.

480 - 485 We found ... layover): these are all quite standard results: the use of multitemporal, geomorphological filters, multi geometry solution is quite present in the current literature, so, I suggest to say that the study confirms that...

Ok, we will modify this sentence and add a reference for Adriano et al., 2020.

485 - 487 The combined effect of stacking hundreds of images with both geometries is an improvement from previous SAR backscatter intensity change studies that have focused on individual acquisition geometries and a relatively small number of SAR images: this is eventually true when the method proposed here is applied (and actually one test case is not significant from a statistical point of view), there is no evidence that other methods proposed in literature would have worked better with long temporal series. furthermore, as far as I can understand, in most of the scenarios only 2 weeks acquisitions, pre-and post images were used.

Rather than state it is an improvement compared to previous studies, we will modify the sentence to "The combined effect of stacking hundreds of images with both geometries can improve the ability to detect landslides". Also note, that the 2 week limitation was only for post-event images. We always used >100 pre-event images. The total number of images used for every single study is listed in the text and in the figure captions.

495 pixel resolution: I guess spatial resolution. I suggest taking into account that in GRD products multilooking is present...

We outlined the pixel resolution in the next part of the sentence.

495 - 496 This resolution limits our ability to detect small landslides
with lengths or widths < 20 m and as a result larger landslides are more likely to be detected: consider that the use of pixel-based applications is not recommended in particular when filtering is not applied, so the assumption of selecting such a small limit for landslide detection is very risky.

We agree. Examination of a single pixel is not recommended, and we emphasize that these data are better suited for larger landslides. We will add some additional text to the revised manuscript that says "Furthermore, users should exercise caution interpreting individual pixels that are isolated in space".

5.3 Challenges with Rapid Response Landslide Detection 5.3.1 Identifying the AOI and EOI

This paragraph seems to at least partially contradict the premises of the strategies: how does, in the end, the method help?

This section is intended to outline the challenges and uncertainties. We do not believe this contradicts the premise, which is that we provide best strategies to detect landslides based on the available data and tools in GEE.

5.3.2 Challenges using the Landslide Heatmap

In this paragraph and in general, in the paper, the importance of false negatives is really underestimated.

While we agree that the issue of false negatives is important, we are not able to directly assess false negatives with our landslide heatmap. We will add a sentence that says "We are not able to

directly assess the number of false negatives with the heatmap because this approach is not used to map individual landslide polygons."

5.3.3 Computation Limits in GEE
5.4 Satellite Acquisition Frequency and Landslide Detection

610 612 Our findings suggest that if the satellite revisit was twice the current revisit time, the modeled AUC score would be ~0.7 just 1 week after the EOI, while if the revisit time was half the current revisit the modeled AUC score would be ~0.65 (red lines in Fig. 11b).: quite arguable: the processes intercepted by changing the frequency sampling and temporal windows can be different so as the final results.

We thank Referee #2 for their feedback. We will add an additional sentence that states "While this finding suggests that, in general, more frequently acquired SAR data will lead to improved ability to detect surface change, the level of improvement may change depending on site specific influences."

5.5 Future Work 6 Conclusions

---

## Author Response (AR2)

**The author's response is shown below in blue text.**

**Comments to the author**:
This is 2nd revision, and I checked it myself as the other reviewer declined to provide his/her review. The paper looks good! I have only minor corrections

**We thank the editor for their review of our manuscript!**

1- I would suggest changing the title to: strategies for rapid landslide heatmap detection ..... and do the necessary changes in the abstract, introduction and conclusion in this regard. I suggest this change as for detecting real landslide objects the authors would need to do further development to include image segmentation in their codes, which is missing in the current manuscript

**We have changed the title to " Generating landslide density heatmaps for rapid detection using open-access satellite radar data in Google Earth Engine" and made changes to the abstract, intro, and conclusion.**

2- The landslide inventory for the 2021 Haiti earthquake has been released now (See https://pubs.er.usgs.gov/publication/ofr20211112?s=09). I would recommend including that in the paper and do the statistical analysis for the comparison instead of visual comparison

**We have updated Fig. 9a, the text, and all relevant links/citations to include the recently published USGS landslide inventory for Haiti (Martinez et al., 2021). However, it is not straightforward to provide a statistical comparison with the USGS inventory due to differences in the final products. The main challenge is that the USGS landslide inventory provides a single point location for each landslide, while our heatmap is generated from numerous points, and as a result a single landslide has many points. We believe the best way to make this comparison is visually, in particular, because our landslide heatmap approach is meant to be a visual method for rapidly detecting areas with high landslide density.**

3- Fig. 7:The USGS landslide inventory (black dots) have not been shown in the Figure

**There is no USGS inventory for Fig. 7, which shows the Huong Phung, Vietnam event. We assume the editor was referring to Fig. 9a, which does show the black dots for the USGS landslide inventory. We have also updated the inventory to include the recently published version by Martinez et al. (2021). Please let us know if there is any issue visualizing the inventory.**

**Lastly, we note that we changed the author order to better reflect the contributions to the manuscript and updated Pukar Amatya's affiliation.**